**Investigation**

# Clusters of acidic and hydrophobic residues can predict acidic transcriptional activation domains from protein sequence

Sanjana R. Kotha,[1,2] Max Valentín Staller[1,2,3,*]

[1]Department of Molecular and Cell Biology, University of California, Berkeley, CA 94720, USA
[2]Center for Computational Biology, University of California, Berkeley, CA 94720, USA
[3]Chan Zuckerberg Biohub—San Francisco, San Francisco, CA 94158, USA

*Corresponding author: Center for Computational Biology, University of California, 16 Barker Hall, Berkeley, CA 94720, USA. Email: mstaller@berkeley.edu

## Abstract

Transcription factors activate gene expression in development, homeostasis, and stress with DNA binding domains and activation domains. Although there exist excellent computational models for predicting DNA binding domains from protein sequence, models for predicting activation domains from protein sequence have lagged, particularly in metazoans. We recently developed a simple and accurate predictor of acidic activation domains on human transcription factors. Here, we show how the accuracy of this human predictor arises from the clustering of aromatic, leucine, and acidic residues, which together are necessary for acidic activation domain function. When we combine our predictor with the predictions of convolutional neural network (CNN) models trained in yeast, the intersection is more accurate than individual models, emphasizing that each approach carries orthogonal information. We synthesize these findings into a new set of activation domain predictions on human transcription factors.

Keywords: transcription, transcription factor, activation domain, transactivation domain, transcriptional activation domain, protein function prediction, convolutional neural network

## Introduction

Transcription factors regulate gene expression with DNA binding domains and effector domains. DNA binding domains are structured, conserved, and recognize related DNA sequences (Latchman 2008; Ferrie et al. 2022; Staller 2022). Profile hidden Markov models can accurately predict DNA binding domains from protein sequence (Stormo 2013; Finn et al. 2016; El-Gebali et al. 2019). Effector domains include repression domains that bind corepressors and activation domains that bind coactivators. Some repression domains can be predicted from protein sequence, and many contain short linear motifs (Tycko et al. 2020; Soto et al. 2022; DelRosso et al. 2023). Activation domains are intrinsically disordered, poorly conserved, and bind to structurally diverse coactivators: these features have made it difficult to predict activation domains from protein sequence (Liu et al. 2006; Dyson and Wright 2016). There are profile hidden Markov models for individual activation domains (e.g. p53 or Hif1a), which can identify activation domains on closely related paralogs or orthologs in other vertebrate species (El-Gebali et al. 2019), but these models are rarely generalizable to predict activation domains on other transcription factors. The ability to predict activation domains from protein sequence would open the door to automated annotation of proteomes and lay a foundation for prioritizing disease-causing mutations. Recently developed models trained in yeast have provided a useful starting point (Ravarani et al. 2018; Erijman et al. 2020; Sanborn et al. 2021). Here, we explore how to improve these models to work on human transcription factors.

Over the last few years, we and others have resolved the key sequence features of strong acidic activation domains, the largest known class (Arnold et al. 2018; Ravarani et al. 2018; Staller et al. 2018, 2022; Erijman et al. 2020; Tycko et al. 2020; Broyles et al. 2021; Sanborn et al. 2021; DelRosso et al. 2023). Based on these sequence features, we proposed an *acidic exposure model* for activation domain function (Staller et al. 2018, 2022). In our acidic exposure model, the hydrophobic residues make key contacts with coactivators but left alone, the hydrophobic residues interact with each other and prevent binding to partners (Fig. 1a). Interspersed between hydrophobic residues are acidic residues that repel each other and keep the hydrophobic residues exposed to solvent. The critical parameter in the acidic exposure model is the balance between acidic and hydrophobic residues. There are cases where acidic residues make fast, low-affinity contacts with basic residues on coactivators (Hermann et al. 2001; Ferreira et al. 2005; Kim and Chung 2020), but these are secondary to hydrophobic contacts.

The acidic exposure model motivated a simple and accurate mechanistic predictor of activation domains (Staller et al. 2022). We investigated if the clustering of acidic, aromatic, and leucine residues could predict activation domains. In our rational

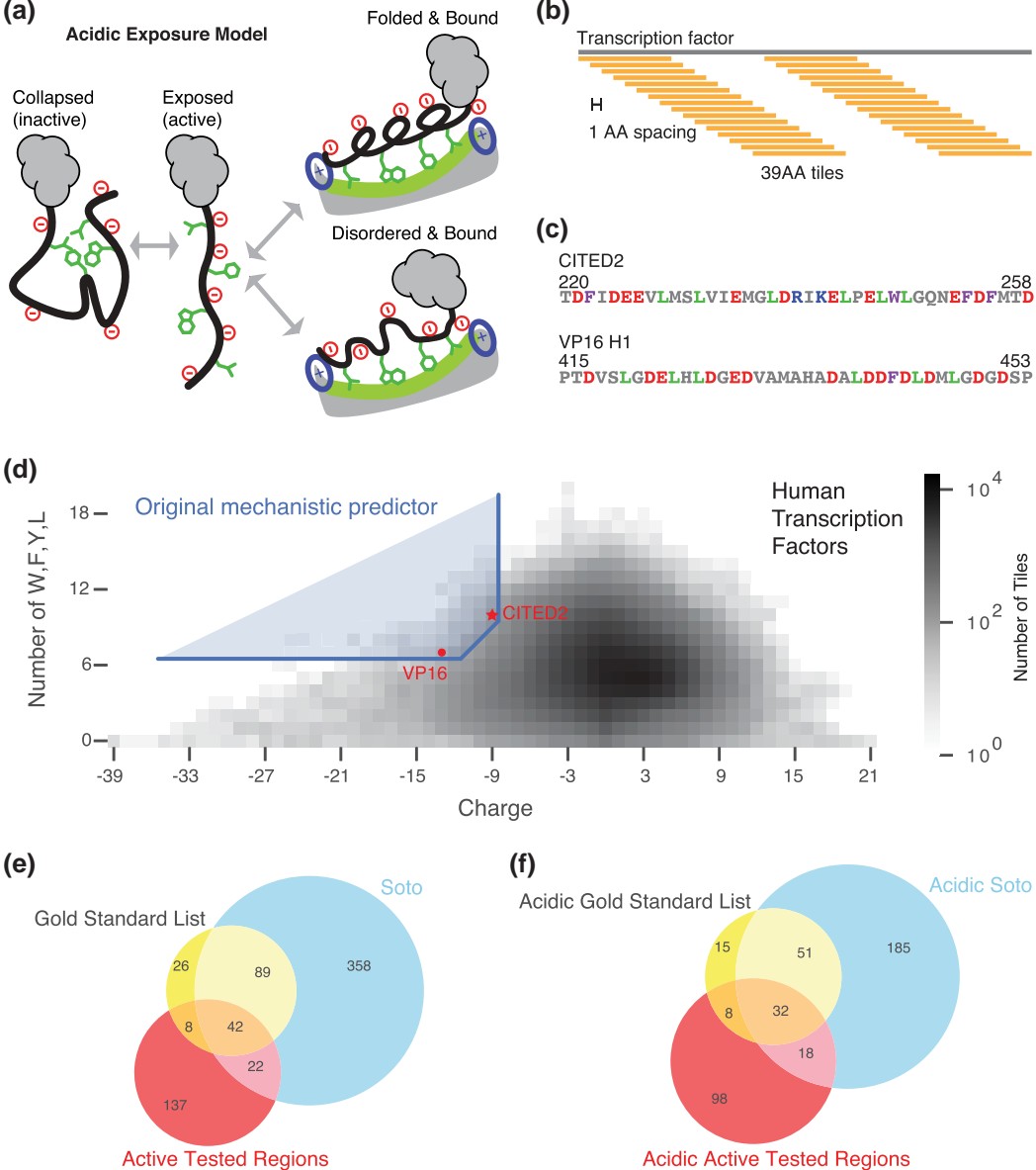

**Fig. 1.** The acidic exposure model can predict acidic activation domains on human transcription factors. a) In the acidic exposure model, intrinsically disordered activation domains dynamically morph between a collapsed, inactive state and an expanded, active state where the key aromatic, and leucine residues are available to interact with hydrophobic surfaces of coactivators. Many activation domains show coupled folding and binding, but can also exhibit fuzzy binding, or remain disordered when bound. b) Schematic for computationally chopping transcription factors into 39-AA tiles spaced every 1-AA. b) Sequences of a 39-AA version of CITED2 and a 39-AA version of VP16 H1 used in the predictor. d) The original mechanistic predictor, the boundary is anchored by CITED2 (star) and VP16 H1 (circle). e) Overlap between the GSL, the Soto list, and the tested regions from Staller et al. (2022). The tested regions include predictions, known activation domains, and negative control regions and had detectable activity in the experiment. f) Overlap between the acidic subsets of the GSL, the Soto list, and tested regions.

mutagenesis experiments, we found that the key sequence features controlling the activity of CITED2 (220–258) and VP16 H1 (415–453) were net negative charge and the number of aromatic and leucine residues (Staller et al. 2022). VP16 H1 is a classic strong acidic activation domain from the human herpes simplex virus and a workhorse in synthetic biology applications (Sadowski et al. 1988; Cress and Triezenberg 1991). CITED2 shuts down the hypoxia response by outcompeting Hif1a for binding to the Taz1 domain of CREB-binding protein (CBP)/p300 (Freedman et al. 2003; Berlow et al. 2017). Both of these very strong activation domains hail from proteins that lack a DNA binding domain. Our original mechanistic predictor showed that regions that resemble CITED2 and VP16 H1 in acidity and the number of aromatic and

leucine residues were enriched for known and new activation domains (Staller et al. 2022). First, we computationally decomposed 1,608 human transcription factors into 39 amino acid (39-AA) tiles spaced every 1-AA, yielding 881 K tiles (Fig. 1b) (Lambert et al. 2018). We used 39-AA tiles because that was the length of the region used in our experiments. We looked for 39-AA tiles that were similar to VP16 H1 and CITED2 (Fig. 1c), using the the formula:

$$(\text{Charge} \leq -9) \text{ AND } (W + F + Y + L \geq 7) \text{ AND}$$
$$(((\text{Charge} + 9) - (W + F + Y + L - 10)) \leq 0), \quad (1)$$

Where W is tryptophan, F is phenylalanine, Y is tyrosine, and L is

leucine. Charge = K + R − D − E. This predictor is anchored by CITED2 (net charge = −9, W + F + Y + L = 10) and VP16 H1 (net charge = −13, W + F + Y + L = 7) (Fig. 1d). The first term selected regions that are at least as acidic as CITED2 (Fig. 1d, vertical threshold). The second term selected regions with at least as many W, F, Y, or L residues as VP16 H1 (Fig. 1d, horizontal threshold). The third term interpolates between these 2 anchor points with a diagonal line of slope 1 (Fig. 1d, diagonal line). In this initial formulation, the feature of the acidic exposure model tested by the predictor is the tendency of aromatic, leucine, and acidic residues to reside together in 39-AA regions. In the human proteome, 1139 39-AA tiles met these criteria, which combined into 144 predicted activation domains. Twenty-six of these predictions overlapped known activation domains, more than expected by chance ($P <$ 1e-5 in permutation tests). We split the longest predictions and tested 150 regions in our high-throughput assay (Staller et al. 2022). Of the 149 recovered fragments, 108 (72%) had detectable activity and 58 (39%) had high activity. This fraction was compared favorably to length-matched random regions and positive controls. The high-activity sequences included 28 known activation domains and 30 new activation domains (Staller et al. 2022). The success of this predictor supported the acidic exposure model in so far as the combination of acidic and W, F, Y, and L residues lead to acidic activation domain function.

While we were developing our mechanistic predictor for human activation domains, 2 CNNs for predicting activation domains in yeast were published. The first, ADpred, was trained on 3.6 million 30-AA random peptides (Erijman et al. 2020). The second, Predictor of Activation Domains using Deep Learning in Eukaryotes (PADDLE), was trained on 53-AA regions that tiled across ~180 *Saccharomyces cerevisiae* transcription factors ($n =$ 10,537 tiles) (Sanborn et al. 2021). Both of these datasets see the same primary signal we saw in our rational mutagenesis: strong activation domains are enriched for acidic and aromatic residues and depleted of basic residues. In yeast, 4 groups have reported the same ranking of amino acid contributions to activity: W > F > Y > L (Ravarani et al. 2018; Staller et al. 2018; Erijman et al. 2020; Broyles et al. 2021; Sanborn et al. 2021). Multiple groups have reported that the absence of positively charged residues is important for activity in yeast (Ravarani et al. 2018; Erijman et al. 2020; Broyles et al. 2021; Sanborn et al. 2021). Sanborn *et al.* reported a correlation between White–Wimbly hydrophobicity and activity, but in this hydrophobicity table the top entries are W > F > Y > L, so the White–Wimbly hydrophobicity scale is emphasizing the most important amino acids (Sanborn et al. 2021). Erijman *et al.* found that [D, E][W, F, Y] dipeptides were enriched in active fragments (Erijman et al. 2020), consistent with our evidence that acidic residues made larger contributions to activity when they were close to hydrophobic residues (Staller et al. 2018, 2022). Individual acidic residues can often (but not always) be removed with little consequence, and instead, acidic residues collectively contribute to activity by creating a permissive context (Staller et al. 2018; Sanborn et al. 2021). All of these signals support the acidic exposure model and are consistent with our mechanistic predictor.

In our previous work, we performed very little optimization of our mechanistic predictor (Staller et al. 2022). Here, we sought to understand why this mechanistic predictor accurately predicted transcriptional activation domains on human transcription factors and to improve its predictive power. We assumed that we were at a local maxima and it would be straightforward to find a much better predictor, the global maximum. Here, we added and removed residues, changed the length-scale, and added grammar. Instead, we found it hard to improve upon the original mechanistic predictor. We found that phenylalanine and leucine make the largest contributions to predictive power, while tryptophan, tyrosine, and methionine contribute modestly. Changing the tile length did not improve model performance. We had previously argued based on experiments that acidic residues make larger contributions to activity when they are close to hydrophobic residues (Staller et al. 2022). We attempted to add this idea to the predictor, but we could not find a consistent, statistically significant signature of this effect. We had more success varying the boundaries of the predictor, finding regions that made small contributions to predictive power, and promising new regions. In parallel, we attempted to predict glutamine-rich, proline-rich, or serine-rich activation domains but were not successful.

Based on our improved understanding of how the original mechanistic predictor functions, we developed a modestly improved mechanistic predictor. Our new predictor uses tryptophan, phenylalanine, and leucine residues, 39-AA windows, and more relaxed thresholds for net charge and hydrophobic residue counts. The new model predicted many more activation domains with minimal loss of accuracy. The primary value of the mechanistic predictor is its simplicity. This model is so simple that we remain surprised this rule was not uncovered earlier. When we compare our mechanistic predictors to CNN models trained in yeast, we found that the intersection is more predictive than individual models, emphasizing that each approach carries distinct information. We synthesize these findings into a new set of activation domain predictions.

## Materials and methods
### Data sources
The gold standard list (GSL) consists of activation domains from 2 sources. The first source is transcription factors with activation domains that were annotated in UniProt, which are in the "data/UniprotActivationDomains_HighqualitySet.csv" file. The second source is activation domains manually curated from the literature, which are in "/data/ActivationDomainsHuman.csv." In "/notebooks/Building the GSL.ipynb," we combined activation domains from both sources into one list, merging entries with the same UniProt ID with overlapping start and end positions. In cases where the annotated boundaries differed, we used the longest region.

For the tested regions in Figs. 1 and 4, we used the activity data for fragments that we previously published (Staller et al. 2022). There were 443 designed fragments of different lengths, but not all were recovered in the assay. In addition, there were multiple cases where the predicted activation domain region was different from the UniProt region, and we experimentally tested both. Here we combined overlapping regions, yielding a total of 356 regions, which include positive controls, negative controls, and predicted activation domains.

The sequences of all 1,608 human transcription factors were downloaded from UniProt (Lambert et al. 2018). We downloaded the sequences of the full human proteome from UniProt. For analysis, we used the protfasta (Holehouse 2021), localcider (Holehouse et al. 2017), metapredict2 (Emenecker et al. 2022), pandas, math, itertools, matplotlib, seaborn, numpy, and scipy python packages.

### Tiling
Each transcription factor sequence is computationally chopped into 39-AA regions spaced every 1-AA (e.g. 1–39, 2–40, 3–41, etc.),

yielding 881K tiles. The full proteome and GSL were similarly decomposed into tiles. The composition of each tile was computed by counting amino acids. The net charge was calculated as R + K − D − E. The original predictor counted W + F + Y + L residues. Other variants of the predictor counted other subsets of residues. For the composition histograms in Fig. 2, we used tiles. To identify the enrichment of specific amino acids in activation domains, we performed a student's *t*-test between the 2 population means. We report all enrichments that were significant at *P* < 0.01 after a Bonferroni correction for multiple hypothesis testing. We used a threshold of 15% for enrichment of individual amino acids (e.g. Q, P, S, or A). This 15% threshold is an emerging standard in the field (DelRosso et al. 2023). We classified activation domains with a net charge less than −3 to be acidic.

## Varying the predictor

The main predictor function is make_predictions() in "notebooks/ AD_predictor_tools.py." The make_predictions() function has parameters that can be used to vary the predictor. We varied the predictor in one way at a time. We adjusted the parameter UpperCorner_slope1 to change the slope of the boundary line originating from the upper corner, CITED2. To create region D in Fig. 4a, we set UpperCorner_slope1 equal to infinity to draw a vertical line to test the predictor with a vertical boundary line instead of a diagonal line. To vary the acidity threshold, we adjusted the parameter upper_corner_c from −13 to 0 to vary the charge of the boundary. To vary the hydrophobic residue count, we adjusted the parameter lower_corner_h from 0 to 15 to vary the count of hydrophobic and aromatic AAs. To vary the composition of AAs used on the y-axis we substituted each pair and triad of AAs for the composition parameter.

We identified all the tiles that satisfied equation (1) and then aggregated the overlapping tiles to predict activation domains. For longer predictions, there can be tiles in the middle that do not individually satisfy equation (1). As a result, when we project the tiles from our tested predictions (Fig. 4b), some of the tiles are outside the prediction region. We initially allowed overlaps

of ≥1-AA, but in practice, the minimum observed overlap was 26 residues. For each version of the predictor, we recalculated the properties of all tiles, adjusted the VP16 and CITED2 anchors, changed the inequalities, found new sets of tiles, aggregated the tiles into predictions, and compared the predictions to the GSL.

We then performed permutation testing. We used the function compare_to_random() in "notebooks/AD_predictor_tools.py" to compare the overlap with the GSL of predictions vs random sequences. To create a random set of sequences, we counted the number of unique UniProt IDs in the predictions and randomly sampled the same number of transcription factors from all human transcription factors. We found that preserving the fact that sometimes there were multiple predictions per transcription factor was more stringent in the permutation test than assuming all predictions were independent. Then, for each unique UniProt ID in the predictions, we recorded the length distribution and number of predictions. We iterated through unique UniProt IDs and randomly selected parts of each sampled transcription factor so that the selected parts had the same number and length distribution as predictions with one UniProt ID. Finally, we recorded the number of times a randomly sampled transcription factor region had any overlap with a GSL entry, using start, end, and UniProt ID to compare. Here we used 1-AA overlaps to increase the stringency of the permutation test. We compared this number to overlaps of predictions with the GSL. We repeated this random sampling for 10,000 permutations. We never randomly sampled as many overlaps as the predictor.

We searched for signatures of molecular grammar in multiple ways. Generally, we would compute parameters for all the activation domains on the GSL and look for enrichment compared to all Lambert transcription factors. For the charge mixture parameters, we tried the original formulation of Kappa and Omega as well as a custom variation of Omega that quantified the mixture of WFYL residues with acidic residues (Das and Pappu 2013; Martin et al. 2016; Ginell and Holehouse 2020). We then compared the parameter between the 2 sets. We also used the measured activities (Staller et al. 2022) to correlate activity with these

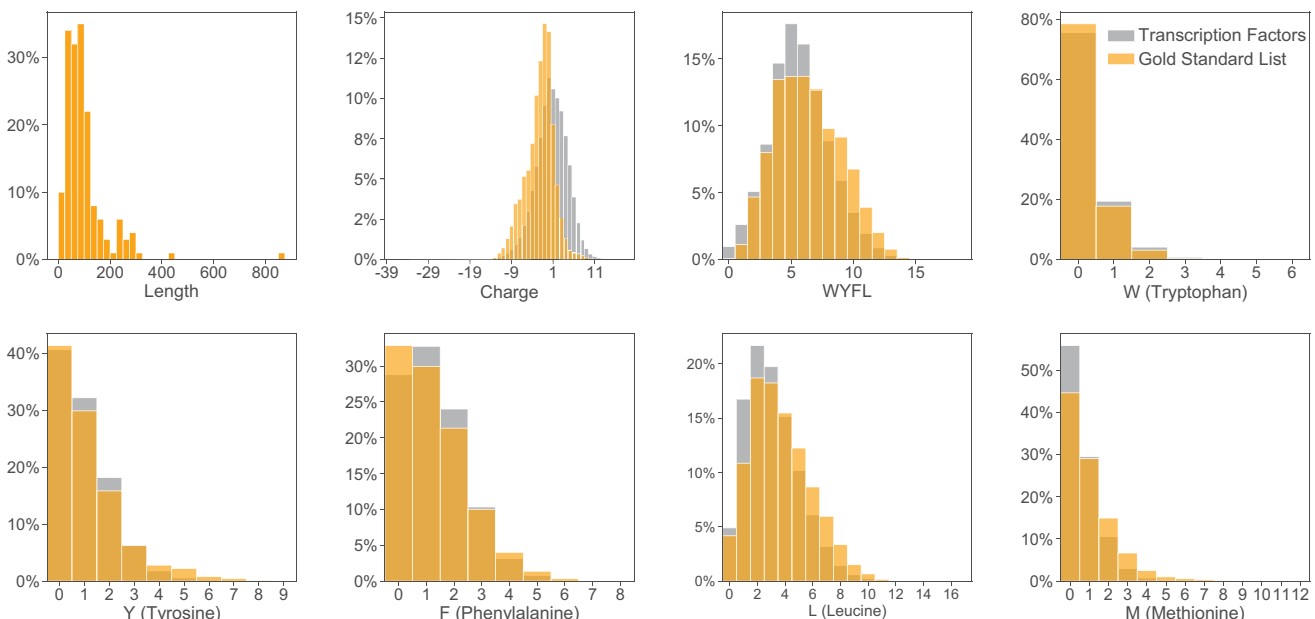

**Fig. 2.** Sequence features of activation domains. Activation domains on the GSL (orange) show very moderate enrichment of individual sequence properties or amino acids compared to the background of transcription factor sequences (gray).

parameters, but the correlations were poor. For the runs of acidic residues, we used the activity measurements for all the tested predictions (Supplementary Fig. 8 in Supplementary File 1).

## Comparing predictions to convolutional neural networks

The 2 neural networks to which we compared the predictor were ADPred and PADDLE (Erijman et al. 2020; Sanborn et al. 2021). We installed ADPred from https://github.com/FredHutch/adpred and then ran it on all Lambert transcription factors. Using the criteria detailed by the authors at https://adpred.fredhutch.org/, we considered sequences with at least 10 consecutive positions with a score of at least 0.8 to be a predicted activation domain. We downloaded the activation domains predicted on human transcription factors by PADDLE at https://cdn.elifesciences.org/articles/68068/elife-68068-fig 3-data1-v3.xlsx. We used predictions of both medium and high strength.

We used the compare_two_predictors() function in "notebooks/AD_comparison_tools.py" to compare our predictor to the neural net predictors. This function first individually compares the overlap of our predictions and a neural net's predictions with a list of known activation domains (either the gold standard or the Soto et al. list). Then, it compares our predictions that overlap with a prediction made by a neural network to the list of known activation domains.

The predicted activation domain regions are deposited at https://zenodo.org/badge/latestdoi/548126430.

The analysis code was deposited at https://zenodo.org/badge/latestdoi/548126430.

## Assessing sensitivity and specificity

To assess the sensitivity and specificity of the revised mechanistic predictor, we first combined the GSL and the Soto list. We again merged overlapping entries by taking the union of coordinates (earlier start and later end). We removed all entries not on the Lambert list of transcription factors (Lambert et al. 2018). Nicole DelRosso provided a list of activation domains from DelRusso et al. (2023). We used a net charge less than –3 as a threshold for acidic activation domains. We considered activation domains that were more than half the length of their transcription factor to be long and excluded them from line 5 of Table 7. We calculated the overlap between predictions and these lists of activation domains as above. Predictions that overlap list entries are labeled as true positives. True negatives are members of the list missed by the predictor. False positives are predictions not present on the reference list, but likely include novel predictions. We calculated sensitivity, specificity, and the f-measure according to the standard conventions.

## Results

To quantify the performance of our activation domain predictors, we curated a GSL of 167 activation domains from 135 proteins, including 129 human transcription factors (Supplementary Table 1). This list combined UniProt annotated activation domains (Downloaded June 2020), individual activation domains curated from NMR papers, and a classic hand-curated list of activation domains (Choi et al. 2000). Overlapping entries were combined by taking the lower start and greater end to make longer annotations. Activation domain boundaries remain difficult to define, so we chose permissive boundaries. When looking for overlaps between lists of activation domains, we started with a very permissive threshold, $\geq 1$

**Table 1.** Types of activation domains on the GSL and Soto et al. list.

| Activation domain subclass | Number of GSL entries | Number of Soto entries |
|---|---|---|
| Acidic | 105 | 290 |
| Glutamine-rich | 12 | 18 |
| Proline-rich | 30 | 119 |
| Serine-rich | 37 | 124 |
| Alanine-rich | 7 | 9 |

overlapping residue, but the minimum observed overlap was 26 residues. We developed our predictors using this GSL and validated the predictors with another recently published "Soto list" (Soto et al. 2022). To avoid circular reasoning, the validation set did not include the 30 novel activation domains correctly identified by the original predictor (Staller et al. 2022). These 30 continue to be correctly identified by our modified predictors.

Our GSL and the Soto list are highly overlapping (Fig. 1e), and acidic activation domains are the most common type (Table 1). The Soto list contains more long activation domains that have not been experimentally minimized. Our GSL is likely enriched for short acidic activation domains that fold into amphipathic alpha helices upon binding coactivators because this mechanism is well represented in the NMR literature. Folding and binding are not essential for activation domain function (Qin et al. 2003; Risør et al. 2021). It is important to note that the entries in the GSL and Soto lists are of variable quality. The activation domains were identified by many labs using many different assays: some are very strong, others very weak, some might be cell-type-specific, and others may yet prove to be false positives. Critically, neither list represents a complete list of true positives, which makes evaluating prediction performance difficult (discussed further below).

Documented activation domains are more diverse than the traditional categories of acidic, proline-rich, or glutamine-rich (Sigler 1988; Gerber et al. 1994; Latchman 2008). There have been scattered references to alanine-rich, glycine-rich, and serine-rich activation domains in the literature, but they have not been recognized as archetypes (Schaeffer et al. 1999; Alerasool et al. 2022; Soto et al. 2022). After exploring several thresholds, we chose 15% as a threshold for composition bias (Supplementary Fig. 1 in Supplementary File 1). Using this 15% threshold, our GSL contains 105 (62.9%) acidic (net charge $< -3$), 12 (7.19%) glutamine-rich (Q-rich), and 30 (18.0%) proline-rich (P-rich) activation domains. In addition, there are 37 (22.2%) serine-rich (S-rich) and 7 alanine-rich (4.19%) (Table 1). The Soto et al. list contains 290 (79.6%) acidic, 18 (3.5%) Q-rich, 119 (22.9%) P-rich, 124 (23.8%) S-rich, 36 (6.9%) glycine-rich, and 9 (1.7%) alanine-rich activation domains. Using our criteria, some activation domains are enriched for more than 1 amino acid, notably acidity, and serines. Annotated acidic activation domains on the 2 lists also overlap with our tested regions (Fig. 1f). Activation domains are enriched for disorder-promoting residues, consistent with the evidence that nearly all activation domains are intrinsically disordered (Liu et al. 2006; Hahn and Young 2011; Oldfield and Dunker 2014; van der Lee et al. 2014). We confirmed that >90% of the activation domains on both lists are predicted to be intrinsically disordered by Metapredict2 (Supplementary Fig. 2 in Supplementary File 1, Emenecker et al. 2022).

We examined the sequence features of our GSL of activation domains. As a background distribution, we used a published list

of 1,608 human transcription factors (Lambert et al. 2018). Annotated activation domains have a wide distribution of lengths because only some have been experimentally minimized (Fig. 2). To make our analyses more consistent, we performed all composition analysis by decomposing each activation domain into all possible 39-AA sliding windows, spaced at 1-AA intervals (e.g. a 45 residue activation domain region would become 7 39-AA tiles, Fig. 1b). We started with 39-AA tiles because that was the length-scale of the original predictor. These 39-AA tiles accommodate activation domains of different lengths and avoid the difficult problem of defining activation domain boundaries. Many activation domains are 39AA or shorter and many long ones contain highly active subregions. Throughout this work, we will analyze proteins by decomposing them into 39-AA tiles and comparing the features of these sets of tiles. Compared to human transcription factor tiles, activation domain tiles show a modest enrichment of net negative charge and M, D, S, L, P, Q, Y, A, V, G residues (t-test, $P < 1e-4$, Bonferroni corrected, Fig. 2, Supplementary Fig. 1 in Supplementary File 1). Activation domains from the GSL do not exhibit extreme properties compared to the background sequence properties of human transcription factors (Supplementary Figs. 1 and 3 in Supplementary File 1). This similarity to the background distribution explains in part why activation domains have been difficult to predict from the sequence.

We next examined how our mechanistic predictor accurately identified acidic activation domains using the combination of net charge and the number of W + F + Y + L residues. We had previously shown this predictor works (Staller et al. 2022), but now we sought to understand why it works. To establish a background distribution, we first examined the sequence features of the full human proteome (Fig. 3a) and 1,608 human transcription factors (Lambert et al. 2018; Fig. 3b). Compared to the full human proteome, transcription factor tiles are slightly positively charged, likely because DNA binding domains contain basic residues that electrostatically interact with the acidic phosphate backbone (Fig. 3d). Transcription factors contain fewer W + F + Y + L residues than the full proteome, likely because they are depleted for transmembrane domains and folded cores of globular proteins (Fig. 3e). Transcription factors contain strong local biases in net charge (Fig. 3b). The most common tile net charge for transcription factors and the proteome is neutral (Fig. 3e). The distribution of transcription factor tile properties is reasonably representative of the full proteome.

Long runs of acidic amino acids are far more common than runs of basic amino acids (Bigman et al. 2022). There exist 11 tiles from transcription factors with a charge of −39 (D/E runs), spanning residues 258–307 of MYT1, a neural transcription factor (Nielsen et al. 2004). Acidic patches can accelerate transcription factor nuclear search processes (Wang et al. 2023). Conversely, the 6 most positively charged tiles from transcription factors were +21, which spanned residues 1845–1908 of SON, a splicing cofactor that binds DNA (Mattioni et al. 1992). The net charge of proteome tiles is also asymmetric, spanning from −39 to +24. The most acidic patch in the proteome is 50 consecutive acidic residues (−50), but the most basic region is +24 (Bigman et al. 2022). It has been argued that this asymmetry is because long positively charged regions interact nonspecifically with nucleic acids and cause toxicity.

When we compared the sequence properties of tiles of activation domains to tiles of full-length transcription factors, we found that acidic activation domains are more acidic than transcription factors (Fig. 3e) and have a slight enrichment for W + F + Y + L residues (Fig. 3d). However, the enrichment for

the combination of acidity and W + F + Y + L residues is much stronger (Fig. 3c). The most acidic regions of transcription factors are not part of known activation domains. Although the most acidic regions are visible on a log scale (Fig. 3c), they are rare and not visible on a linear scale (Fig. 3e). Similarly, the transcription factor tiles with the most W + F + Y + L residues are not part of activation domains (Fig. 3c, d). No single property distinguishes activation domains, but the combination of acidity and W + F + Y + L residues can enrich a subclass of acidic activation domains, as we have seen before (Staller et al. 2022). Our predictor finds activation domains that balance acidic residues against W + F + Y + L residues (Fig. 3c).

Similarly to how acidic activation domains are not the most acidic regions of transcription factors, P-rich and Q-rich activation domains are not among the transcription factor tiles with the most Ps or Qs (Supplementary Figs. 4 and 5 in Supplementary File 1). This observation is consistent with evidence that Q-rich activation domains contain a lower fraction of Qs than proteins with true poly-Q regions, like Huntingtin (Ruff et al. 2014). The traditional activation domain labels were assigned before the completion of the human genome project, namely, before there was a proper null distribution against which to show enrichment.

In contrast, S-rich activation domains are enriched for serine residues compared to the background of transcription factor sequences (Supplementary Fig. 1, Supplementary 7f in Supplementary File 1). The tiles spanning S-rich activation domains on our GSL had more serine residues than tiles from all other regions of transcription factors (t-test, pval = 1.37e-229). Some activation domains increase activity when they are phosphorylated (Raj and Attardi 2017; De Mol et al. 2018; Peng et al. 2019; Conti et al. 2023), prompting us to search for an enrichment of common phosphorylation motifs (e.g. serine-proline (SP) and serine-glutamine (SQ)). Activation domains and repression domains contain more documented phosphorylation sites than DNA binding domains (Soto et al. 2022). We initially detected an enrichment of short motifs when we compared the GSL to all transcription factor tiles. As a more stringent control, we shuffled the sequences of the S-rich activation domains and counted random occurrences of phosphorylation motifs. Compared to these sequence permutations, phosphorylation motifs did not occur in activation domains more often than expected by chance. We conclude that phosphorylation motifs are not enriched in S-rich human activation domains beyond what is expected by chance.

We tested whether the combination of W + F + Y + L and P, Q, or S could enrich the activation domains of each class, but none of these combinations worked (Supplementary Fig. 4 and 5 and 6 in Supplementary File 1). Our mechanistic predictor of acidic activation domains is not easily extended to other classes. We and others have speculated that P, Q, or S residues could keep hydrophobic motifs exposed to solvent in a manner analogous to acidic residues (Staller et al. 2018; DelRosso et al. 2023), but this analysis and recently published experiments suggest a more complex mechanism is at work.

## Dissecting the mechanistic predictor

In order to improve the mechanistic predictor, we next sought to understand why it worked and which activation domains from the GSL it was identifying. We first asked if human activation domains were more similar to VP16 H1 or the CITED2 by dividing our initial prediction region into 3 triangular regions (Fig. 4a, Table 2). Region A has the highest fraction of correct predictions (12/47).

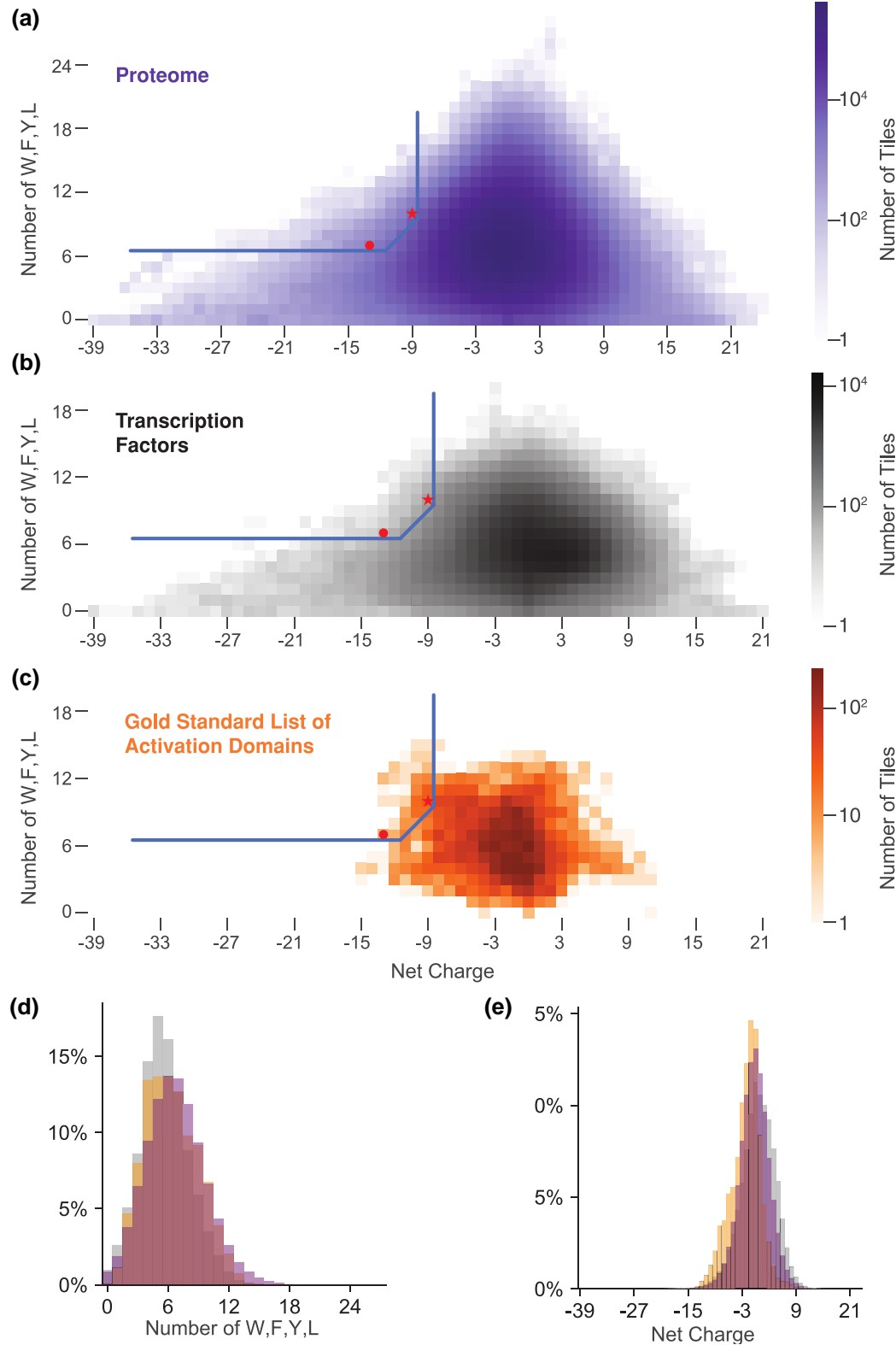

**Fig. 3.** The combination of acidic and WFYL residues predicts activation domains. a) We decomposed the proteome into 39-AA tiles and calculated the net charge and counted WFYL residues for each tile. For each combination of these 2 properties, we counted tiles to create a 2D histogram that is visualized as a heatmap. The full proteome (purple) has a more diverse distribution of tiles than other protein sets we examined. b) Tiles from annotated transcription factors (gray, Lambert list). c) Tiles from the entire GSL of activation domains (orange). d) Overlaps of histograms of tiles with varying numbers of WFYL residues, colors as in a, b, c, and e) The GSL of activation domains (orange) is enriched for acidic tiles.

Region B, where acidic and hydrophobic residues were balanced, had the most correct predictions (25/133). In contrast, Region C identified no activation domains on the GSL (0/18). Tiles like CITED2 and balanced tiles were most likely to be activation domains. This analysis prompted us to remove Region C from the updated predictor.

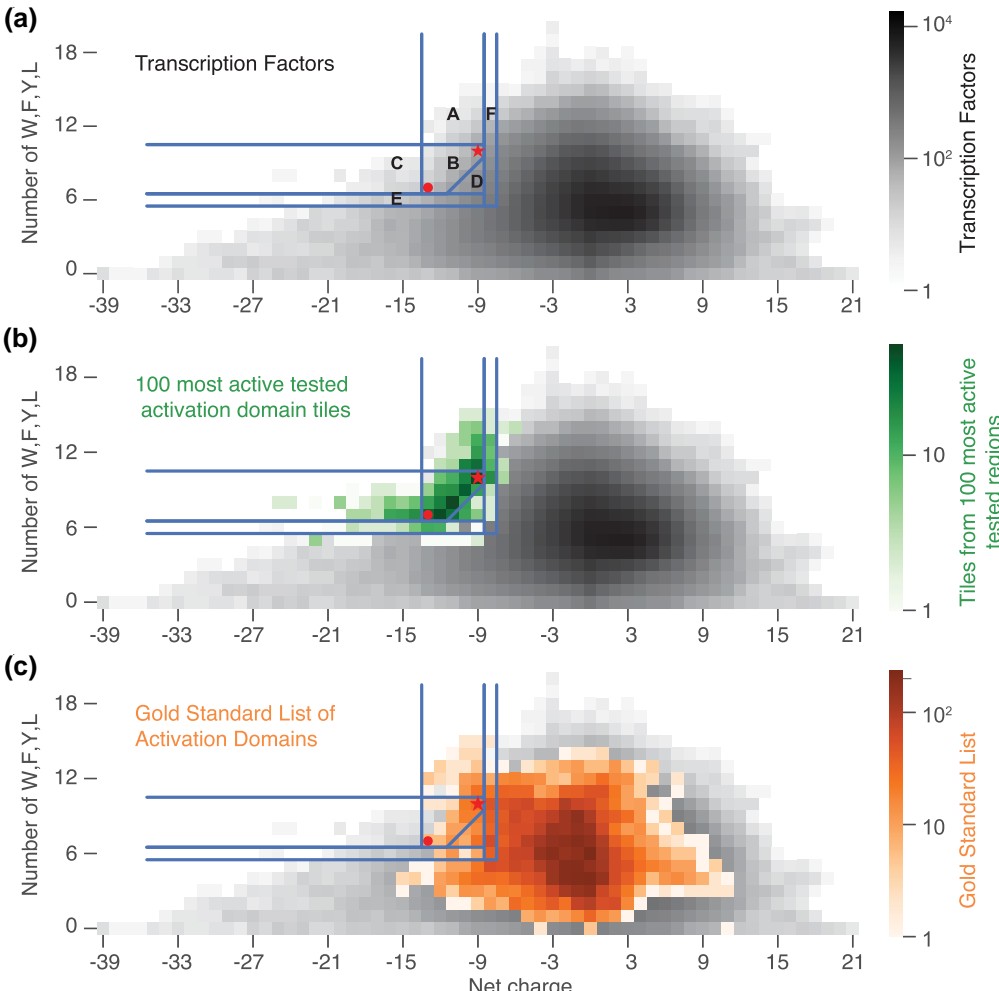

**Fig. 4.** Regions that balance acidity and WFYL residues are most predictive of activation domains. a) We split our original activation domain predictor into regions a, b, and c. We tested the predictive power of additional regions d, e, and f. The dot indicates VP16 H1 and the star indicates CITED2, the anchor points for the original mechanistic predictor. b) We took the 100 strongest activation domains in our assay (Staller et al. 2022) and projected the constituent tiles onto the regions in a. The peak of the tile distribution is in region b. c) Projecting tiles of all members of the GSL over the regions in a.

**Table 2.** Subregions of the mechanistic predictor from Fig. 3a differ in the power to detect members of the GSL of activation domains.

| Region in Fig. 4a | Number of predictions | GSL overlap count | GSL overlap proportion (precision) | Proportion of acidic GSL found (sensitivity) |
|---|---|---|---|---|
| A | 47 | 12 | 0.255 | 0.114 |
| B | 133 | 25 | 0.188 | 0.238 |
| C | 18 | 0 | 0.000 | 0.000 |
| D | 285 | 28 | 0.098 | 0.267 |
| E | 202 | 7 | 0.035 | 0.067 |
| F | 633 | 50 | 0.079 | 0.476 |

Reciprocally, when we took our correct predictions (i.e. predictions with high activity in our experiment) (Staller et al. 2022) and examined their tiles, the peak of distribution lay along the line connecting CITED2 and VP16 (Fig. 4b). Virtually all of the tiles that mapped to Region C came from activation domains that also contained tiles that mapped to Region B. This new analysis of our published data further emphasizes how balance is the key to accurate prediction.

To determine the parameters that contribute most to the predictor, we performed a sensitivity analysis. We removed each of the 8 AAs in the predictor and recomputed predictive power (Supplementary Table 2). F, L, and charge make the largest contributions to sensitivity and specificity, likely because these residues are more common. Despite being enriched in the GSL activation domains, Y made very small contributions. W's make modest contributions to predictive power because they are rare. Similarly, we varied the length of the tiling windows and did not see improvement (Supplementary Table 3). These new variations of the mechanistic predictor revealed there was no simple way to improve upon the original predictor.

For further comparison, we replaced the y-axis parameter with all singles, pairs, and triplets of amino acids (Supplementary Tables 4, 5, and 6). For each combination, we changed the thresholds based on the sequences of CITED2 and VP16 H1 (methods). Leucine was the single amino acid with the highest specificity. Leucine was present in the 5 pairs with the highest specificity and in 11/12 triplets with the highest specificity. Together, this analysis emphasized that a high number of leucine residues is predictive of human activation domains.

**Table 3.** Changing the interpolation between the 2 anchor points, CITED2 and VP16, predicts new activation domains.

| | Number of predictions | GSL overlap count | GSL overlap proportion (precision) | P value (PERMUTATION) |
|---|---|---|---|---|
| Original mechanistic predictor | 144 | 26 | 0.181 | <1e-4 |
| Original mechanistic predictor plus Region D (corner) | 312 | 37 | 0.119 | <1e-4 |

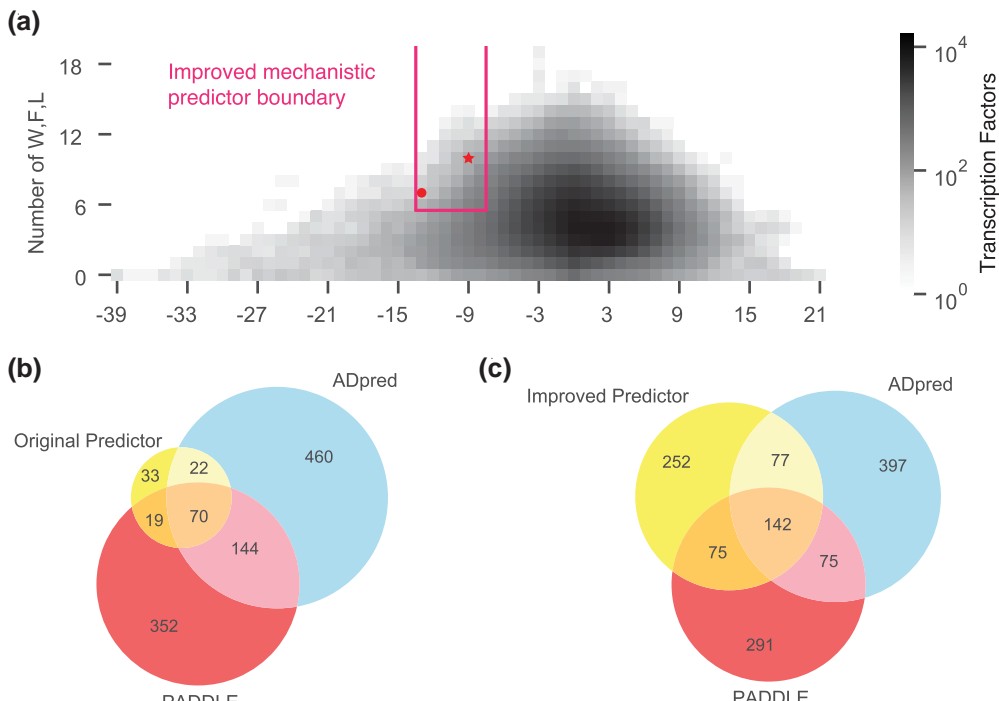

**Fig. 5.** Intersections of the mechanistic predictors with the published neural network models from yeast. a) The region of the improved mechanistic predictor. b) The overlap between the original mechanistic predictor, ADpred, and CBP, IDRs predictions. c) The overlap between the improved mechanistic predictor, ADpred, and PADDLE predictions.

## Expanding the boundaries of the mechanistic predictor

To make new predictions, we altered the boundaries of the mechanistic predictor to include more tiles. We had already tested nearly all the original predictions and found that 72% had detectable activity in our assay (Staller et al. 2022). Making new predictions requires expanding one or more boundaries, even if it comes at the cost of reduced sensitivity. First, we changed the interpolation between VP16 H1 and CITED2 from a diagonal line to a corner, using the minimum value of each activation domain to create a right triangle below CITED2 (Fig. 4a, **Region D**, Table 3) using equation (2).

$$(\text{Charge} \leq -9) \text{ AND } (W + F + Y + L \geq 7). \tag{2}$$

Region D contains 285 predictions, including 168 new predictions and 11 additional activation domains on the GSL (Table 2). Next, we lowered the W + F + Y + L boundary (Fig. 4a, **Region E**) and lowered the acidic boundary (Fig. 4a, **Region F**). When we consider only acidic activation domains, the projected tiles look very similar to those of the full GSL (Supplementary Fig. 7 in Supplementary File 1). Regions D and F contain the most new

predictions. In permutation tests, both versions of the revised predictor continue to detect more activation domains than expected by chance (Table 3).

## An improved mechanistic predictor

We report an improved mechanistic predictor of acidic activation domains (Fig. 5a). The primary improvement of this expanded predictor is that it makes many new predictions with small decreases in accuracy (see below). We selected a trapezoidal region (equation 3), used 7 amino acids (W, F, L, D, E, R, K), and expanded the charge and hydrophobic thresholds by one:

$$(-13 \leq \text{Charge} \leq -8) \text{ AND } (W + F + L \geq 6). \tag{3}$$

The trapezoid better emphasizes the balance between hydrophobic and acidic residues (Fig. 5a). This region predicted 546 activation domains. Of these 546 predictions, 47 are on the GSL, and 51 regions have high activity in our assays (Table 4). Moreover, 104/546 of our predictions are on the Soto list. The improved mechanistic predictor identified 47/105 acidic activation domains on the GSL (44.8% sensitivity) and 104/290 acidic activation domains on the Soto list (35.9% sensitivity). This improved mechanistic

**Table 4.** Comparison of Soto and Gold Standard Lists (GSL).

| | Number of predictions | GSL overlap count | GSL overlap proportion (precision) | Soto overlap count | Soto overlap proportion (precision) |
|---|---|---|---|---|---|
| Original mechanistic predictor | 144 | 26 | 0.181 | 46 | 0.3194 |
| Improved mechanistic predictor | 546 | 47 | 0.086 | 104 | 0.19 |

**Table 5.** Intersection of the mechanistic predictor and convolutional neural network models improves prediction accuracy.

| Predictor | Predictions | GSL overlap count | Percent of predictions on GSL (precision) | Soto overlap count | Percent of predictions on Soto (precision) |
|---|---|---|---|---|---|
| Original mechanistic predictor | 144 | 26 | 18.06% | 46 | 31.94% |
| ADPred | 721 | 53 | 7.35% | 138 | 19.14% |
| Original mechanistic predictor ∩ ADPred | 87 | 23 | 26.44% | 40 | 45.98% |
| PADDLE | 602 | 76 | 12.62% | 167 | 27.74% |
| Original mechanistic predictor ∩ PADDLE | 89 | 25 | 28.09% | 44 | 49.44% |

predictor makes 406 new predictions on 342 transcription factors. These transcription factors hail from a diverse set of families including many nuclear hormone receptors, Sox, Klf, and Zinc finger transcription factors.

## Sequence grammar

We attempted to improve our predictor by adding sequence grammar, which we define as the arrangement of amino acids. Examples of strict grammar include short linear sequence motifs (SLiMs), where amino acids must have a defined spacing or arrangement, e.g. $\Phi xx\Phi\Phi$, where $\Phi$ is a bulky hydrophobic residue (Warfield et al. 2014; Dyson and Wright 2016). Examples of weak grammar include cases where acidic residues make larger contributions to activity when they are close to hydrophobic residues, or "mini motifs" of one acidic residue followed by an aromatic residue ([D or E][W or F or Y], represented as the regular expression [DE][WFY]) that contribute to activity (Ravarani et al. 2018; Staller et al. 2018; Erijman et al. 2020). In addition to searching for motifs, we looked for amphipathic alpha helices, distance dependencies between the aromatic and acidic residues, dipeptides, the *Kappa* charge mixture parameter (Das and Pappu 2013), the *Omega* charge and proline mixture parameter (Martin et al. 2016; Ginell and Holehouse 2020), and repetitive runs of amino acids. The only statistically significant grammar signal was that tiles with long runs of acidic residues were less likely to be activation domains (Supplementary Fig. 8 in Supplementary File 1). Other studies have argued there is little to no grammar in activation domains (Erijman et al. 2020; Sanborn et al. 2021; DelRosso et al. 2023). The grammar that does exist is highly degenerate and flexible, making it hard to detect with our small sample size. Ultimately, we did not add grammar to the mechanistic predictor.

## Combining the mechanistic predictor with neural networks improves performance

We found that combining our mechanistic predictor with CNN predictors trained on yeast activation domains improved predictive power beyond the performance of either alone. Intersecting our predictor ($n = 144$) and PADDLE ($n = 604$) increased sensitivity (Fig. 5b). For the 89 activation domains predicted by both models, 25 (28.1%) were on the GSL and 44 (49.4%) were on the Soto list (Table 5, Supplementary Table 7). In addition, 88 had been tested in our activation domain assay and 45 (51.1%) had activity (Staller et al. 2022). This result implies each predictor brings orthogonal

information. The 60 predictions removed by this intersection have many runs of acidic residues, consistent with the grammar analysis above.

We found similar predictive improvement when we intersected our mechanistic predictor with ADpred. ADpred made 721 predictions on human transcription factors. Twenty-seven of these are on the GSL and 45 overlap with the Soto list (Table 5). Intersecting the ADpred predictions with our predictor led to 87 overlaps: 23 (26.4%) with the GSL and 40 (46.0%) with the Soto list (Table 5). We tested 86 of these regions in our experiments, and 40 (46.5%) had detectable activity in our assay (Staller et al. 2022). The intersection once again was more accurate than either model alone. ADpred and PADDLE scores are correlated (Fig. 5b). We conclude that combining the CNNs and our mechanistic predictor yields the most accurate predictions.

Intersecting this revised predictor with the CNNs yielded 139 high-confidence predictions (Fig. 5c, Table 6, Supplementary Table 8). We anticipate that testing this new set of predictions will identify new activation domains.

Notably, there are 5 true activation domains found by our original predictor that are thrown out by PADDLE and ADpred. These activation domains from FOS, TIGD7, ZN513, TIGDF, and ZN777 contain many leucines (>10%), which is interesting because leucines make larger contributions to activity in human activation domains than in yeast activation domains (Staller et al. 2022). Indeed, based on our 15% threshold, FOS and ZN513 qualify as leucine-rich regions. We hypothesize that these leucine-rich activation domains are a metazoan innovation that binds to activation domain binding domains not present in yeast, such as the TAZ1 and TAZ2 domains of CBP/p300.

Why does the combination of the mechanistic predictor and the CNNs improve performance? Some of this improvement is likely because each approach contributes orthogonal information. We also believe that the overlap might be providing some insight into how the CNNs work. Convolutional neural networks are black-box models, which makes it difficult to understand the source of their accuracy. ADpred and PADDLE take as inputs primary sequence, predicted secondary structure, and predicted intrinsic disorder, but the models do not tell us which feature, or combination of features, is most important for prediction. For regulatory DNA CNNs, there are emerging tools for extracting mechanistic insight (Avsec et al. 2021), but analogous tools for interpreting protein sequence models remain limited (Erijman et al. 2020;

**Table 6.** Intersection of the improved mechanistic predictor and convolutional neural network models improves prediction accuracy.

| Predictor | Predictions | GSL overlap count | Percent of predictions on GSL (precision) | Soto overlap count | Percent of predictions on Soto (precision) |
|---|---|---|---|---|---|
| Improved mechanistic predictor | 546 | 47 | 8.61% | 104 | 19.05% |
| ADPred | 721 | 53 | 7.35% | 138 | 19.14% |
| Improved mechanistic predictor ∩ ADPred | 216 | 35 | 16.20% | 74 | 34.26% |
| PADDLE | 602 | 76 | 12.62% | 167 | 27.74% |
| Improved mechanistic predictor ∩ PADDLE | 217 | 44 | 20.28% | 86 | 39.63% |

**Table 7.** Performance of the revised mechanistic predictor.

| Benchmark list | Predictions | Benchmark | True positives | False positives | False negatives | Positive predictive value (precision) | True positive rate (sensitivity) | F-score |
|---|---|---|---|---|---|---|---|---|
| All GSL + Soto | 546 | 519 | 110 | 436 | 409 | 0.201 | 0.212 | 0.207 |
| Acidic GSL + Soto | 546 | 294 | 105 | 441 | 189 | 0.192 | 0.357 | 0.250 |
| All GSL + Soto limited transcription factors | 159 | 519 | 110 | 49 | 409 | 0.692 | 0.212 | 0.324 |
| Acidic GSL + Soto limited transcription factors | 143 | 294 | 105 | 38 | 189 | 0.734 | 0.357 | 0.481 |
| Acidic GSL + Soto limited transcription factors, without long activation domains | 139 | 260 | 101 | 38 | 159 | 0.727 | 0.388 | 0.506 |
| All DelRosso | 546 | 242 | 94 | 452 | 148 | 0.172 | 0.388 | 0.239 |
| Acidic DelRosso | 546 | 203 | 94 | 452 | 109 | 0.172 | 0.463 | 0.251 |
| Acidic DelRosso limited transcription factors | 132 | 203 | 94 | 38 | 109 | 0.712 | 0.463 | 0.561 |

Mahatma et al. 2023). We speculate that the overlap between the mechanistic predictor and the CNNs suggests that composition plays a substantial role in their performance.

## Assessing the performance of the predictors

Assessing the positive predictive value (precision) and the true positive rate (recall) of the mechanistic predictor is challenging because existing lists of activation domains are incomplete, making it difficult to evaluate which predictions are false positives and which are novel predictions. We had previously used permutation tests to randomly select transcription factor regions to show the mechanistic predictor identified more activation domains than expected by chance (Staller et al. 2022). All variations of our predictors continue to meet this threshold (Methods). Here, we further assessed the predictive power of the improved mechanistic predictor (546 predictions) in multiple ways. First, we assumed the combined GSL and Soto lists (541 entries) represented the full set of human activation domains. Based on this assumption, there are 110 true positives, the positive predictive value is 0.201, and the true positive rate is 0.203, indicative of poor performance (Table 7). Second, we used only the acidic members of these lists. Under this condition, the positive predictive value is 0.194, and the true positive rate is 0.311. We consider these estimates the minimum performance of our predictor. To estimate the maximum performance of our predictor, we next limited our assessment to the 127 transcription factors with at least one entry on the GSL or Soto lists, assuming that all activation domains on these transcription factors are known. Based on this assumption, the positive predictive value is 0.692 and the true positive rate remains 0.203, indicating an increase in performance. Repeating this limited assessment on acidic activation domains, the positive predictive value is 0.667, and the true positive rate is 0.311. Fifth, we removed entries that comprised more than half the transcription factor ($n = 47$) because in these cases little or no experimental

effort was devoted to finding a minimal activation domain. Here, the positive predictive value is 0.706 and the true positive rate is 0.344. This alternative assessment with a smaller search space likely represents the maximum performance of our model. The true performance of our revised mechanistic predictor sits between these 2 estimates.

During the review process, a systematic screen for human activation domains in K562 cells was published (DelRosso et al. 2023). This screen examined transcription factors and chromatin regulators, so we only looked at the transcription factors. Using this list of activation domains from transcription factors, we repeated the above analyses and obtained similar performance as assessed by the positive predictive value, the true positive rate, and the F-score (Table 7). Together, these analyses give us confidence that the mechanistic predictor can find activation domains on human transcription factors.

Finally, as previously published, the most rigorous assessment of our predictor is experimental validation (Staller et al. 2022). When we tested the 144 predictions from the mechanistic predictor, 72% had detectable activity. This precision of 0.72 is comparable to the maximum of our estimates above (Table 7). The PADDLE CNN achieves a similar level of precision, 70%, in the recent screen for human activation domains (DelRosso et al. 2023). This CNN performance is comparable to our published success rate (Staller et al. 2022) and the newly calculated estimates above (Table 7). Together, these analyses show the improved mechanistic predictor identifies many more candidate activation domains with minimal loss of accuracy.

Next, we estimated true negatives called by the predictor in several ways. First, we looked for transcription factors with no known activation domains and no predicted activation domains and found 915/1,231 (74.3%) correctly predicted as true negatives. Second, we looked at transcription factors with repression domains as defined by Soto and found that 66/384 (17%) had

predicted activation domains, indicating a low false positive rate. Alternatively, these may be bifunctional transcription factors that activate and repress transcription. Third, we looked at the number of predicted activation domains that overlap repression domains and found 33 examples. Of these, 27/33 overlap KRAB domains, which are a very interesting set of false positives. We had previously tested 13 predictions that overlap KRAB repression domains (Staller et al. 2022). One is the KRAB domain of Zn473, which is one of the 4 highly divergent KRAB domains that function as activation domains (Tycko et al. 2020). This Zn473 KRAB domain behaved as an activation domain in our experiments (Staller et al. 2022). In a few cases, the prediction covers the full KRAB (e.g. Znf12), but in most cases, the prediction overlaps the N-terminal region, which is least important for repression activity (Tycko et al. 2020). Three of these predictions are highly active in our experiments (Zn473, Zn561, and Zn571), 5 have detectable activity, and 5 have very low activity, as expected for repression domains. Given that KRAB domains appear to convert to activation domains at a low rate on long evolutionary time scales, we might be catching some of these regions in transition: the full-length KRAB domain is still a repression domain, but the N-terminal half is becoming a weak activation domain. Overall, we conclude that the mechanistic predictor has a high true negative rate and a low false positive rate.

Based on these assessments, the mechanistic predictor is accurate and sensitive. However, we wish to emphasize that the main utility of the mechanistic predictor is its simplicity and interpretability.

## The predictors identify one subclass of acidic activation domains

Both our original mechanistic predictor and the improved mechanistic predictor do not identify all of the acidic activation domains on the gold standard and Soto lists. Similarly, neither ADpred (53/105) nor PADDLE (76/105) can detect all these acidic activation domains. We have tuned the mechanistic predictor to have a low false positive rate at the expense of a high false negative rate. While there are multiple interpretations of this result, we favor the interpretation that there are multiple subclasses of acidic activation domains and that the existing predictors can find the one subclass that is well described by the acidic exposure model. These models miss activation domains where activity is regulated by modifying the net charge with post-translational modifications. For example, the first activation domain of p53 (net charge = −6, WFYL = 8) has three sites that increase activity when phosphorylated (S15, T18, S20, net charge = −12, WFYL = 8) (Raj and Attardi 2017). These residues are interspersed with key aromatic and leucine residues consistent with the acidic exposure model. In this case, the resting sequence falls outside the activation domain predictor boundary (equation 3), but the activated, phosphorylated state crosses it over the boundary. Understanding how phosphorylation controls activation domain function is an exciting area of future inquiry.

## Discussion

Accurate computational models for predicting activation domains from protein sequence will advance basic science and precision medicine. Computationally annotating activation domains would allow studies of how paralogous transcription factors diversify after duplication and enable evolutionary comparisons of domain shuffling. Comprehensive lists of transcription factors

with activation domains could improve gene regulatory networks by adding signs to the connections inferred from genome binding data (Hummel et al. 2023) or by distinguishing direct and indirect connections inferred from genetic perturbations. Predicting activation domains is a key step toward building models that predict how mutations in activation domains modulate activity, which, in the long term, could classify patient mutations in activation domains as benign or pathogenic (Richards et al. 2015; Starita et al. 2017). These classifications could group patients for the development of targeted therapies or prioritize variants for base-editing gene therapies.

Our mechanistic predictor is valuable because it is simple and interpretable. Its accuracy comes from the acidic exposure model, which describes a subclass of acidic activation domains that balance acidic residues with key hydrophobic residues. The analyses presented here confirm and strengthen our previous conjectures (Staller et al. 2022). This work explains why the predictor works. The predictor's success further supports one critical feature of the acidic exposure model, that hydrophobic motifs require an acidic context. Our acidic exposure model is related to the stickers and spacers model for how specialized intrinsically disordered regions form condensates (Martin et al. 2020), albeit with a more active role for the spacers. So far, we can predict only acidic activation domains. Analogous predictors of P-rich, Q-rich, or S-rich activation domains do not work (Supplementary Fig. 4 and 5 and 6 in Supplementary File 1).

Why are so many activation domains negatively charged? What is the mechanistic advantage of acidity? This question has been repeated many times since it was posed by Paul Sigler (Sigler 1988). In principle, exposure to hydrophobic residues could be achieved by positively charged residues, but, in practice, positively charged residues inhibit activation domain function (Ravarani et al. 2018; Erijman et al. 2020; Broyles et al. 2021). Many coactivators have positively charged surfaces, and long-range, low-affinity fast electrostatic interactions have been documented (Hermann et al. 2001; Ferreira et al. 2005). These electrostatic interactions can be important for making activation domain coactivator interactions diffusion-limited in "fly-fishing" models of activation domain coactivator interactions (Kim et al. 2018; Kim and Chung 2020). We believe there are advantages to acidic activation domains and disadvantages to basic activation domains. The first advantage is that acidic residues electrostatically repel the DNA, allowing the activation domain to stick out and catch coactivators. Second, acidic activation domains can have low-affinity electrostatic intramolecular interactions with basic DNA binding domains, which can increase the specificity of DNA binding via competitive inhibition (Stott et al. 2014; Krois et al. 2018; He et al. 2019; Wang et al. 2023). Third, acidity makes it possible to post-translationally regulate activation domain activity with phosphorylation (Conti et al. 2023). We see 2 potential disadvantages to basic activation domains: first, nonspecific, electrostatic binding to DNA that could compete with coactivator binding or inhibit nuclear search. Second, cation-π interactions between basic residues and aromatic residues (e.g. arginine–tyrosine interactions) could drive collapse (or condensate formation) and make positively charged residues less effective at keeping some aromatic residues exposed to solvent (Wang et al. 2018). There are also examples of positively charged repression domains (Soto et al. 2022; DelRosso et al. 2023). Together, these observations explain why so many activation domains are acidic.

Activation domains display very flexible sequence grammar. If activation domains had strict sequence grammar requirements for function, we would have seen these signatures in the

evolutionary record, mutagenesis, or in tiling experiments. An early grammar model, the 9aaTAD model, can identify known activation domains, but in high-throughput screens of random peptides or yeast transcription factors, it does not detect more often than expected by chance (Piskacek et al. 2007; Erijman et al. 2020; Sanborn et al. 2021). Instead, we see evidence for very flexible grammar or no grammar. The evidence for no grammar is that random peptides can have activation domain activity and that shuffling activation domain sequence can preserve or even sometimes increase activity (Ma and Ptashne 1987; Arnold et al. 2018; Ravarani et al. 2018; Staller et al. 2018; Erijman et al. 2020; Sanborn et al. 2021). The high accuracy of our grammar-less composition-based predictor supports both a no-grammar model and a flexible-grammar model. The evidence against no-grammar models is that shuffling activation domain sequence can both increase and decrease activity (Staller et al. 2018; Sanborn et al. 2021). Loss of activity is more common when shuffling disrupts an alpha helix (Sanborn et al. 2021; Staller et al. 2022). In these shuffle mutants, the arrangement of amino acids, i.e. the grammar, is modulating activity, ruling out a strict no-grammar model. We can rule out a strict-grammar model and we can rule out a no-grammar model, so we are left with a very flexible-grammar model.

How do we square a very flexible-grammar with the documented role of short linear motifs and amphipathic alpha helices? The dominant model for activation domains is that they are anchored by a hydrophobic short linear motif embedded in a permissive context. At this time, the features of the context are more clearly defined than the motifs. The context is acidic residues and intrinsic disorder. In some cases, the motifs are clearly present, conserved, and contribute to activation domain activity (Dyson and Wright 2016). A motif in an amphipathic alpha helix is a very effective way to coherently display several hydrophobic residues to a coactivator (Giniger and Ptashne 1987). Amphipathic alpha helices are a good solution for building an activation domain (Dyson and Wright 2016), but, critically, they are not the *only* solution. Motifs are uncommon and rarely generalize beyond a few transcription factors. Surveys of random peptides, yeast transcription factors, and human transcription factors found enrichment of only [DE][WFY] "mini motifs" (Arnold et al. 2018; Ravarani et al. 2018; Erijman et al. 2020; Broyles et al. 2021; Sanborn et al. 2021; DelRosso et al. 2023). We argue that the critical distinction is that a motif or an amphipathic helix is not the only way for a cluster of hydrophobic residues to interact with a coactivator–many arrangements are functional. A growing number of fuzzy interactions have been documented, but they are likely underreported because of investigator bias and a higher burden of proof (Brzovic et al. 2011; Warfield et al. 2014; Tuttle et al. 2018; Risør et al. 2021). Fuzzy binding is consistent with a highly flexible-grammar.

It is not clear at what point the motif ends and the context begins. In our mutagenesis of VP16 and CITED2, we found that virtually every hydrophobic residue contributed to activity, blurring the distinction between motifs and context. Adding aromatic residues near a motif—in essence extending the motif—increases activation domain activity (Warfield et al. 2014; Staller et al. 2018). Based on mutagenesis of Abf1, one group has argued that motif quality and context quality both contribute to function and that each can compensate for the other (Langstein-Skora et al. 2022). Sequences that contain many functional elements will be composition driven and grammar-independent (e.g. DWDWDWDWDWDWDWDWDWDWDW (Ravarani et al. 2018)). Grammar and motifs will be important on the margin for sequences that barely have the right composition to be functional but can function when the residues are appropriately arranged into a motif. Regions with fewer acidic and hydrophobic residues will likely be more reliant on grammar. Critically, even in a highly flexible-grammar regime, not all arrangements of residues will be active. Real sequences are likely to be on this margin because neutral drift is likely pulling strong activators down to the minimum functional level maintained by negative selection. Marginally active activation domains would be easier to regulate by post-translational modifications. Weak or regulated activation domains allow more precise combinatorial control of gene expression. We speculate that there is no boundary between motifs and context.

## Conclusion

We conclude that the human proteome contains a class of strong acidic activation domains that can be recognized by the clustering of W, F, L, and acidic residues. This balance between acidity and hydrophobicity accurately predicts known activation domains, many of which are well described by our acidic exposure model. Our work implies there are other classes of acidic activation domains that are not predicted by our model and which likely bind to other coactivators. Forthcoming, comprehensive maps of activation domains (DelRosso et al. 2023) will create the opportunity to test and improve the mechanistic predictors and the next generation of CNN models (Mahatma et al. 2023). Our analyses emphasize the need to characterize the sequence features that control activity of Q-rich, S-rich, and P-rich activation domains.

## Data availability

Our hand-curated, GSL of activation domains is in Supplementary Table 1. All the data and code is available in the Github repository: https://zenodo.org/badge/latestdoi/548126430. The activity data from measuring the predictions of the original mechanistic predictor are from Supplementary Table 4 of Staller et al. 2022. The PADDLE activation domain predictions were taken from a supplementary table of Sanborn et al. 2021. The Soto list of activation domains is a supplemental table from Soto et al. 2022. DelRosso activation domain data is from DelRosso et al. 2023. The protein sequences of the Lambert transcription factors were from UniProt. Supplementary Table 9 contains our combined list of published and experimentally identified activation domains.

Supplemental material available at GENETICS online.

## Acknowledgments

The authors thank Jordan Stefani, Zeba Wunderlich, Alex Holehouse, Vinson Fan, and Darren Kahan for their comments on the manuscript. Thanks to Shiyi Yang and Angelica Lam for help in setting up and running ADpred. Thank you to Nicole DelRosso for providing a list of activation domains. Thank you also to the reviewers for very useful feedback on the manuscript.

## Funding

S. R. K. is an undergraduate at UC Berkeley supported by the STEM Excellence through Equity & Diversity (SEED) Scholars Honors Program. This work was supported by the Burroughs Wellcome Fund Postdoctoral Enrichment Program, Simons Foundation grant 1018719, and USA National Science Foundation grant 2112057. M.V.S is a Chan Zuckerberg Biohub—San Francisco Investigator.

## Conflicts of interest

The author(s) declare no conflict of interest.

## Author contributions

S. R. K. Conceptualization, Software, Formal analysis, Writing. M.V.S. Conceptualization, Methodology, Writing.

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

*Editor: C. Kaplan*