## [Peer Review File · Genetics]

Clusters of acidic and hydrophobic residues can predict acidic transcriptional activation domains from protein sequence

Sanjana Kotha and Max Staller

NOTE: The reviews and decision letters are unedited and appear as submitted by the reviewers.

In extremely rare instances and as determined by a Senior Editor or the EIC, portions of a review may be redacted. If a review is signed, the reviewer has agreed to no longer remain anonymous.

The review history appears in chronological order.

Review Timeline:

Submission Date:	2023-02-25
Editorial Decision:	2023-03-27
Resubmission Recieved:	2023-05-18
Editorial Decision:	2023-06-13
Resubmission Received:	2023-06-16
Editorial Decision:	2023-06-26
Revision Received:	2023-06-27
Accepted:	2023-07-03

March 26, 2023

GENETICS-2023-305986

The balance of acidic and hydrophobic residues predicts acidic transcriptional activation domains from protein sequence

Dear Dr. Staller:

Three experts in the field have reviewed your manuscript, and I have read it as well. While your manuscript is not currently acceptable for publication in GENETICS, we would welcome a substantially revised manuscript. The reviewers have comments and concerns to be addressed in a revised manuscript. You can read their reviews at the end of this email.

The reviews represent a range of opinions about the work. Reviewer 1 has some concerns about statistical analysis of predictor effectiveness. Additionally, there are concerns about a clear delineation of the novelty of the work and the impact of the results. Reviewers 2 and 3 were more positive but one theme across reviews was concern about need for a more clear delineation of what is new here versus Staller 2022. This is important so the reader can distinguish the prior work from these new studies . We look forward to receiving your revised manuscript. Please let the editorial office know approximately how long you expect to need for revisions.

Upon resubmission, please include:

1. A clean version of your manuscript;
2. A marked version of your manuscript in which you highlight significant revisions carried out in response to the major points raised by the editor/reviewers (track changes is acceptable if preferred);
3. A detailed response to the editor's/reviewers' feedback and to the concerns listed above. Please reference line numbers in this response to aid the editor and reviewers.

Your paper will likely be sent back out for review.

Additionally, please ensure that your resubmission is formatted for GENETICS
<https://academic.oup.com/genetics/pages/general-instructions>

Follow this link to submit the revised manuscript: <https://genetics.msubmit.net/cgi-bin/main.plex?el=A1NR3FQd2A2VZR1I6A9ftdKMPNZvclqHbfb0UK1mYOXwZ>

Sincerely,

Craig Kaplan
Associate Editor
GENETICS

Approved by:
Karen Arndt
Senior Editor
GENETICS

Reviewer #1 (Comments for the Authors (Required)):

Overall, this is a follow-up on their previous paper that used a simple equation to predict Acidic Activation Domains. The authors present an appealing mechanistic model based on amphipathic helices (but they don't apparently use this model, see below).

Prediction of activation domains (and more generally, understanding how they function) is a topic of general interest. However, in my view, the contributions of this study provide limited practical benefit and limited biological insight.

On the practical side, (i.e., the potential benefits of their predictor) the authors use non-standard methodology terminology to evaluate their predictor, which confuses the reader. The presentation and evaluation falls far below the current standard in bioinformatics. For example, they report "Proportion overlapping the Soto list", which I believe is (a lower bound on) positive predictive power or precision (true positives/predicted positives). The authors' "Improved model" shows a decrease (30-50%) in this critical metric. Hence we are not convinced that this improved model is an improvement. Of course, positive predictive power can only be meaningfully compared at an equivalent true positive rate (or recall), which it is not clear if they have done. Alternatively, there are standard approaches that combine the two. Without use of standard comparison metrics, it is very difficult for the reader to understand what has been achieved here. The authors write:

"The true positive rate of the original (26/105) and improved predictors (47/105) are, strictly speaking, not amazing."

It is unclear if the improvement in true positive rate is actually expected due to the reduction in positive predictive power, or what would constitute, "strictly speaking amazing" vs. "strictly speaking not amazing" in this context. Claims of improvement of predictive power need to be supported with statistical analysis of performance metrics.

The major conclusion is that their model when used with the machine learning methods improves predictive power. However, it is not clear that their "improved" method (Table 6) combined with deep learning is better than their "original" method when combined with deep learning (Table 4). These comparisons should be done using standard metrics and statistical support/significance for the improvement must be provided (e.g., error bars on the positive predictive power). These are all standard practices in the field, and as it stands this paper falls far below of the standard practice.

On the other hand, the biological insight that can be gleaned from the work here is also limited. They refer to their model as a "mechanistic" model of "acidic exposure model" based on patterning of residues in acidic exposure on helices that form when activation domains are bound. However, they show later that patterns of residues does not appear to influence the predictions, and then don't use helical patterning in their predictions. The authors claim that they "could not detect statistically significant grammar signals", but they not clear about whether they explicitly tested amphipathic helices or other patterns relating to the acidic exposure model. At least, this needs to be explained fully, or the comments about "mechanistic predictors" should be removed. The authors don't discuss why the deep learning predictors are adding to the predictive power. In contrast to the authors' arguments, the CNNs could be capturing the amphipathic helices, and therefore they would be the "mechanistic" models.

There are many minor issues:

-To me the interesting question they could address is: what could they be adding that the PADDLE CNN doesn't have (it has secondary structure, it has grammar/motifs, it has IDR predictions)? As far

as I can tell this is not discussed.

-it's not explained what the notation [DE][WFY] means, but I interpret this to be a regular expression as in ELM

-the two activation domains they refer to (CITED2 and VP16H1) should be introduced for readers that are not familiar.

-In all the figures with the gold standard TFs, it's not clear if they are showing only the acidic ones or all of them.

Reviewer #2 (Comments for the Authors (Required)):

This is an interesting study which directly expands upon prior works related to characterization of transcriptional activation potential of acidic activation domains and development of predictor of such function based on protein characteristics. In the current study, the authors were able to improve their previous predictor which now performs on par with published neural network models (Erijman et al., 2020, Sanborn et al., 2021). The intersection of the revised predictor with each of the neural network models significantly improves the accuracy over individual approaches. Other key findings include: 1) Available models are most likely able to predict only a specific sub-class of acidic activation domains (ADs); 2) Observation about the existence of leucine-rich activation domains which are found in metazoans but not in yeast).; 3) Acknowledging that the often-overlooked contribution of post-translational modifications (most importantly phosphorylation) to AD function likely confounds the results obtained with available tools. The findings are generally convincing but some of the conclusions were already made in previous works (see below). Nevertheless, I believe this work provides significant advance in our understanding of the mechanisms involved in transcription activation and will be of interest to other researchers.

Major comments:

- Some of the conclusions are repetitive with the previous studies (Erijman et al., 2020, Staller et al., 2022), e.g. the apparent finding that the balance between hydrophobic and acidic residues is a feature of strong acidic ADs. These are taken from Staller et al., 2022: "We found that strong ADs balance the number of acidic residues against the number of aromatic and leucine residues" and "We exploited this observation to create an AD predictor that scans for a high but balanced composition of acidity and hydrophobicity". It does not seem that the conclusion changes in the current work (except some details related to the contributions of individual amino acids). I believe that the language should be adjusted to better reflect which findings are novel and which were already reported.

Minor comments:

- Some figures could use refinement. For example, X-axes in Fig. S2 are too dense to read.
- The text has a significant number of spelling errors. Please revise.

Reviewer #3 (Comments for the Authors (Required)):

The authors explore sequence properties of human transcription activation domains using lists of

presumed or known ADs in the context of a prior AD model that was based on amino acid composition. Although the authors new improved model presented here is not strikingly accurate (with significant numbers of false positives and false negatives), I think it is important in that it examines in more detail the sequence properties that most closely track with activity. In a recently published paper, Staller and collaborators proposed that an important feature of ADs is an appropriate balance between acidic and hydrophobic residues. The model makes sense with what is known about how ADs interact with a limited set of targets. In this work, the authors additional studies support this hypothesis and test the limits of how many ADs may fit this model. For example, they show that non acidic ADs don't typically fit the model. I think that a revised manuscript could make an important contribution to the field in guiding the thinking about how to consider properties of proteins that have AD function vs non functional sequences.

- 1) One limitation of this approach seems to be the list of activators used for the analysis. It seems like the "gold standard list" is derived from the literature and presumably contains ADs that have been identified in many different assays by many different investigators. If so, there are certainly some false positives as well as ADs that are very strong and very weak and everything in between. Using the entire list as a single group to develop a predictor seems like an important limitation in the analysis and this should be noted when this list is introduced.
- 2) What's the basis for choosing 15% threshold to detect compositional bias? How would the numbers look if the whole proteome were to be categorized in a similar way (as in Table-1)?
- 3) Net charge of acidic ADs from GSL doesn't look very different from all ADs (Fig. 3C vs Fig. S7B). Doesn't it suggest that 15% threshold is not an ideal criterion for categorizing ADs?
- 4) Fig 1E - why are random negative controls included in the plot? It seems unnecessary to include them here and leaving them out would increase the overlap between the lists.
- 5) Fig S3 should be eliminated as the authors state that this analysis is not useful.
- 6) Pg 8, top: there is no reference to a figure or table with the ser-enriched ADs. In contrast to this statement Fig S6 seems to show no such enrichment.
- 7) Table 2: error in table, value should be 0.098, not .98.
- 8) Table 2 is confusing. I would like to know the fraction of GSL sequences are correctly predicted - not the percent overlap between all predictions and the GSL.
- 9) Why is Fig 4A,B plotted on a different x axis scale compared with 4C?
- 10) I suggest that it would be easier to understand the explanation of new boundaries if it is shifted to the "Dissecting the mechanistic predictor" section.

Kotha and Staller, Response to Reviews, Genetics, Spring 2023

Reviewer #1 (Comments for the Authors (Required)):

Overall, this is a follow-up on their previous paper that used a simple equation to predict Acidic Activation Domains. The authors present an appealing mechanistic model based on amphipathic helices (but they don't apparently use this model, see below).

We stumbled upon the original predictor by accident and performed very little optimization. This paper is our attempt to improve the predictor. We are surprised that the original predictor has been hard to improve. We assumed that we were in the foothills near a mountain and that it would be easy to ascend to higher levels of performance, but instead we found we started near a peak.

We have revised the end of the introduction to state our initial goals, the negative results, and the modest improvement of the predictor. We believe this context will help clarify how this paper builds upon the previous work and which ideas are previously published and which ideas are new.

Prediction of activation domains (and more generally, understanding how they function) is a topic of general interest. However, in my view, the contributions of this study provide limited practical benefit and limited biological insight.

On the practical side, (i.e., the potential benefits of their predictor) the authors use non-standard methodology terminology to evaluate their predictor, which confuses the reader. The presentation and evaluation falls far below the current standard in bioinformatics. For example, they report "Proportion overlapping the Soto list", which I believe is (a lower bound on) positive predictive power or precision (true positives/predicted positives). The authors' "Improved model" shows a decrease (30-50%) in this critical metric. Hence we are not convinced that this improved model is an improvement. Of course, positive predictive power can only be meaningfully compared at an equivalent true positive rate (or recall), which it is not clear if they have done. Alternatively, there are standard approaches that combine the two. Without use of standard comparison metrics, it is very difficult for the reader to understand what has been achieved here. The authors write:

"The true positive rate of the original (26/105) and improved predictors (47/105) are, strictly speaking, not amazing."

It is unclear if the improvement in true positive rate is actually expected due to the reduction in positive predictive power, or what would constitute, "strictly speaking amazing" vs. "strictly speaking not amazing" in this context. Claims of improvement of predictive power need to be supported with statistical analysis of performance metrics.

The major conclusion is that their model when used with the machine learning methods improves predictive power. However, it is not clear that their "improved" method (Table 6)

combined with deep learning is better than their "original" method when combined with deep learning (Table4) These comparisons should be done using standard metrics and statistical support/significance for the improvement must be provided (e.g., error bars on the positive predictive power). These are all standard practices in the field, and as it stands this paper falls far below of the standard practice.

Assessing the true positive and false positive rates of the mechanistic predictor is difficult because all existing lists of activation domains are incomplete. Before submission, we had tried to execute a few variations of this analysis, but we were unsatisfied with them. Unlike traditional bioinformatic models, our predictors do not have thresholds that can be continuously varied to generate ROC or PRC curves. They exist as a single point on an ROC or PRC plot. As a result, it is not possible to compare PPP at equivalent TPR.

In response to your comment, we have reopened this problem and tried more variations. We have added the following section to the manuscript reflecting our new analysis:

(Page 18) **Assessing the performance of the predictors**

Assessing the positive predictive value (precision) and the true positive rate (recall) of the mechanistic predictor is challenging because existing lists of activation domains are incomplete, making it difficult to evaluate which predictions are false positives and which are novel predictions. We had previously used permutation tests to randomly select transcription factor regions to show the mechanistic predictor identified more activation domains than expected by chance (Staller *et al.* 2022). All variations of our predictors continue to meet this threshold (Methods). Here, we further assessed the predictive power of the improved mechanistic predictor (546 predictions) in multiple ways. First, we assumed the combined GSL and Soto lists (541 entries) represented the full set human activation domains. Based on this assumption, there are 110 true positives, the positive predictive value is 0.201, and the true positive rate is 0.203, indicative of poor performance (Table 7). Second, we used only the acidic members of these lists. Under this condition, the positive predictive value is 0.194, and the true positive rate is 0.311. We consider these estimates the minimum performance of our predictor. To estimate the maximum performance of our predictor, we next limited our assessment to the 127 TFs with at least one entry on the GSL or Soto lists, assuming that all ADs on these TFs are known. Based on this assumption, the positive predictive value is 0.692 and the true positive rate remains 0.203, indicating an increase in performance. Repeating this limited assessment on acidic activation domains, the positive predictive value is 0.667, and the true positive rate is 0.311. Fifth, we removed entries that comprised more than half the TF (n=47) because in these cases little or no experimental effort was devoted to finding a minimal AD. Here, the positive predictive value is 0.706 and the true positive rate is 0.344. This alternative assessment with a smaller search space likely represents the maximum performance of our model. The true performance of our revised mechanistic predictor sits between these two estimates.

Table 7: Performance of the revised mechanistic predictor

Benchmark list	Predictions	Benchmark	True Positives	False Positives	False Negatives	Positive Predictive Value (Precision)	True Positive Rate (Sensitivity)	F-Score
All GSL+Soto	546	519	110	436	409	0.201	0.212	0.207
Acidic GSL+Soto	546	294	105	441	189	0.192	0.357	0.250
All GSL+Soto limited TFs	159	519	110	49	409	0.692	0.212	0.324
Acidic GSL+Soto limited TFs	143	294	105	38	189	0.734	0.357	0.481
Acidic GSL+Soto limited TFs, without long ADs	139	260	101	38	159	0.727	0.388	0.506
All DelRosso	546	242	94	452	148	0.172	0.388	0.239
Acidic DelRosso	546	203	94	452	109	0.172	0.463	0.251
Acidic DelRosso limited TFs	132	203	94	38	109	0.712	0.463	0.561

During the review process, a systematic screen for human activation domains in K562 cells was published (DelRosso *et al.* 2023). This screen examined transcription factors and chromatin regulators, so we only looked at the transcription factors. Using this list of activation domains from transcription factors, we repeated the above analyses and obtained similar performance as assessed by the positive predictive value, the true positive rate, and the F-score (Table 7). Together, these analyses give us confidence that the mechanistic predictor can find activation domains on human transcription factors.

Finally, as previously published, the most rigorous assessment of our predictor is experimental validation (Staller *et al.* 2022). When we tested the 144 predictions from the mechanistic predictor, 72% had detectable activity. This high level of precision is comparable to the maximum of our estimates above. The PADDLE CNN achieves a similar level of precision, 70%, in the recent screen for human activation domains (DelRosso *et al.* 2023). This performance is comparable to our published success rate (Staller *et al.* 2022) and the new calculated estimates above. Together, these analyses show the mechanistic predictor is accurate.

Next, we estimated true negatives called by the predictor in several ways. First, we looked for transcription factors with no known ADs and no predicted ADs and found 915/1231 (74.3%) correctly predicted as true negatives. Second, we looked at transcription factors with repression domains as defined by Soto and found that 66/384 (17%) had predicted activation domains, indicating a low false positive rate. Alternatively, these may be bifunctional transcription factors that activate and repress transcription. Third, we looked at the number of predicted ADs that overlap repression domains and found 33 examples. Of these, 27/33 overlap KRAB domains, which are a very interesting set of false positives. We had previously tested 13 predictions that overlap KRAB repression domains (Staller *et al.* 2022). One is the KRAB domain of Zn473, which is one of the four highly divergent KRAB domains that function as activation domains (Tycko *et al.* 2020). This Zn473 KRAB domain behaved as an activation domain in our experiments (Staller *et al.* 2022). In a few cases, the prediction covers the full KRAB (e.g. Znf12), but in most cases the prediction overlaps the N-terminal region, which is least important for repression activity (Tycko *et al.* 2020). Three of these predictions are highly active in our experiments (Zn473, Zn561, and Zn571), five have detectable activity, and five have very low activity, as expected for repression domains. Given that KRAB domains appear to

convert to activation domains at a low rate on long evolutionary time scales, we might be catching some of these regions in transition: the full length KRAB domain is still a repression domain, but the N-terminal half is becoming a weak activation domain. Overall, we conclude that the mechanistic predictor has a high true negative rate and a low false positive rate.

Based on these assessments, the mechanistic predictor is accurate and sensitive. However, we wish to emphasize that the main utility of the mechanistic predictor is its simplicity and interpretability.

On the other hand, the biological insight that can be gleaned from the work here is also limited. They refer to their model as a "mechanistic" model of "acidic exposure model" based on patterning of residues in acidic exposure on helices that form when activation domains are bound. However, they show later that patterns of residues does not appear to influence the predictions, and then don't use helical patterning in their predictions. The authors claim that they "could not detect statistically significant grammar signals", but they not clear about whether they explicitly tested amphipathic helices or other patterns relating to the acidic exposure model. At least, this needs to be explained fully, or the comments about "mechanistic predictors" should be removed. The authors don't discuss why the deep learning predictors are adding to the predictive power. In contrast to the authors' arguments, the CNNs could be capturing the amphipathic helices, and therefore they would be the "mechanistic" models.

We have revised the text to enumerate other grammar signals for which we looked. These signals include published motifs, amphipathic alpha helices, and runs of amino acids. (see page 15).

The feature of the acidic exposure model we are testing is that W,F,L residues need to be surrounded by acidic residues. We have revised the text in multiple places to state this idea more explicitly.

We believe that the role of amphipathic alpha helices in activation domain biology is overstated because of publication bias in the NMR literature: if an activation domain forms a helix when it binds a partner, the result is publishable. If a helix is not detected, the burden of proof for publication is higher, and many of these negative results are not published.

We added a new section of text to discuss the relative contributions of the mechanistic model and the CNNs:

(Page 16) "Why does the combination of the mechanistic predictor improve performance? CNNs are black box models, which makes it difficult to understand the source of their accuracy. ADpred and PADDLE take as inputs primary sequence, predicted secondary structure, and predicted intrinsic disorder, but the models do not tell us which feature, or combination of features, is most important for prediction. For regulatory DNA CNNs, there are emerging tools for extracting mechanistic insight (Avsec *et al.* 2021), but analogous tools for protein sequence have not yet been established. The overlap between the mechanistic predictor and the CNNs suggests that composition, and not grammar, plays a large role in CNN performance."

We are not sure that we fully understood the last sentence of this critique. Are you suggesting that CNNs are mechanistic models? CNNs are black box models that are very difficult to interpret mechanistically. Instead, maybe you are saying that ADpred and PADDLE use alpha helix predictions as inputs and might be giving these input features a high weight when predicting activation domains? This is formally possible but very hard to determine. It remains very difficult to ascertain which input features to a CNN are contributing most to predictive power. For regulatory DNA CNNs, there are emerging tools for extracting mechanistic insight (Avsec *et al.* 2021), but analogous tools for protein sequence have not yet been established. With collaborators, we are developing new CNN models for predicting activation domains from protein sequence (Mahatma *et al.* 2023). We are investigating the contributions of different sequence features to function in that work.

There are many minor issues:

-To me the interesting question they could address is: what could they be adding that the PADDLE CNN doesn't have (it has secondary structure, it has grammar/motifs, it has IDR predictions)? As far as I can tell this is not discussed.

We think the overlap between the mechanistic model and the CNN is telling us that the CNN is more simple than it looks, namely that many of the CNN's predictions can be found by looking at the counts of only eight amino acids. We believe that PADDLE is primarily detecting composition with a minor role for grammar. Proving this conjecture is difficult, but we would be happy to add this speculation to the manuscript.

-it's not explained what the notation [DE][WFY] means, but I interpret this to be a regular expression as in ELM

We have defined the use of this regular expression.

-the two activation domains they refer to (CITED2 and VP16H1) should be introduced for readers that are not familiar.

This is a great suggestion, and we have expanded this part of the introduction.

-In all the figures with the gold standard TFs, it's not clear if they are showing only the acidic ones or all of them.

Thank you for pointing this out. Reviewer 3 raised a similar point. Our plots make it difficult to distinguish between the acidic sequences and all the sequences. In the figure legends, we have attempted to clarify the sets to which we refer.

Reviewer #2 (Comments for the Authors (Required)):

This is an interesting study which directly expands upon prior works related to characterization of transcriptional activation potential of acidic activation domains and development of predictor

of such function based on protein characteristics. In the current study, the authors were able to improve their previous predictor which now performs on par with published neural network models (Erijman et al., 2020, Sanborn et al., 2021). The intersection of the revised predictor with each of the neural network models significantly improves the accuracy over individual approaches. Other key findings include: 1) Available models are most likely able to predict only a specific sub-class of acidic activation domains (ADs); 2) Observation about the existence of leucine-rich activation domains which are found in metazoans but not in yeast); 3) Acknowledging that the often-overlooked contribution of post-translational modifications (most importantly phosphorylation) to AD function likely confounds the results obtained with available tools. The findings are generally convincing but some of the conclusions were already made in previous works (see below). Nevertheless, I believe this work provides significant advance in our understanding of the mechanisms involved in transcription activation and will be of interest to other researchers.

Major comments:

- Some of the conclusions are repetitive with the previous studies (Erijman et al., 2020, Staller et al., 2022), e.g. the apparent finding that the balance between hydrophobic and acidic residues is a feature of strong acidic ADs. These are taken from Staller et al., 2022: "We found that strong ADs balance the number of acidic residues against the number of aromatic and leucine residues" and "We exploited this observation to create an AD predictor that scans for a high but balanced composition of acidity and hydrophobicity". It does not seem that the conclusion changes in the current work (except some details related to the contributions of individual amino acids). I believe that the language should be adjusted to better reflect which findings are novel and which were already reported.

Thank you for this great suggestion. We have revised the text to reflect more clearly which ideas were previously published and which ideas are new. We have also changed the title to: *Clusters of acidic and hydrophobic residues can predict acidic transcriptional activation domains from protein sequence.*

We stumbled upon the original predictor by accident and performed very little optimization. This paper is our attempt to improve the predictor. We are surprised that the original predictor has been hard to improve. We assumed that we were in the foothills near a mountain and that it would be easy to ascend to higher levels of performance, but instead we found we started near a peak.

Minor comments:

- Some figures could use refinement. For example, X-axes in Fig. S2 are too dense to read.

Thank you for pointing this out. Please see the revised Figure S2.

- The text has a significant number of spelling errors. Please revise.

The senior author is mildly dyslexic and terrible at spelling. We hired a graduate student from the English Department to check the manuscript for spelling errors.

Reviewer #3 (Comments for the Authors (Required)):

The authors explore sequence properties of human transcription activation domains using lists of presumed or known ADs in the context of a prior AD model that was based on amino acid composition. Although the authors new improved model presented here is not strikingly accurate (with significant numbers of false positives and false negatives), I think it is important in that it examines in more detail the sequence properties that most closely track with activity. In a recently published paper, Staller and collaborators proposed that an important feature of ADs is an appropriate balance between acidic and hydrophobic residues. The model makes sense with what is known about how ADs interact with a limited set of targets. In this work, the authors additional studies support this hypothesis and test the limits of how many ADs may fit this model. For example, they show that non acidic ADs don't typically fit the model. I think that a revised manuscript could make an important contribution to the field in guiding the thinking about how to consider properties of proteins that have AD function vs non functional sequences.

1) One limitation of this approach seems to be the list of activators used for the analysis. It seems like the "gold standard list" is derived from the literature and presumably contains ADs that have been identified in many different assays by many different investigators. If so, there are certainly some false positives as well as ADs that are very strong and very weak and everything in between. Using the entire list as a single group to develop a predictor seems like an important limitation in the analysis and this should be noted when this list is introduced.

Thank you for this suggestion. We have revised the description of the list with the following paragraph:

(Page 7) "Our gold standard list and the Soto list are highly overlapping (Figure 1E), and acidic activation domains are the most common type (Table 1). The Soto list contains more long activation domains that have not been experimentally minimized. Our gold standard list is likely enriched for short acidic activation domains that fold into amphipathic alpha helices upon binding coactivators because this mechanism is well represented in the NMR literature. Folding and binding is not essential for activation domain function (Qin et al. 2003; Risør et al. 2021). It is important to note that the entries in the gold standard list and Soto lists are of variable quality. The activation domains were identified by many labs using many different assays: some are very strong, others very weak, some might be cell-type-specific, and others may yet prove to be false positives. Critically, neither list represents a complete list of true positives, which makes evaluating prediction performance difficult (discussed further below)."

2) What's the basis for choosing 15% threshold to detect compositional bias? How would the numbers look if the whole proteome were to be categorized in a similar way (as in Table-1)?

We tried multiple thresholds before selecting 15%. As we increase the threshold, fewer sequences meet the threshold and resultant lists are therefore very short and harder to analyze. We have revised the text to state this fact and updated Figure S2 to show the compositional bias of tiles from gold standard list activation domains. Others in the field are also using 15% as a threshold (DeiRosso *et al.* 2023).

(Page 7): “After exploring several thresholds, we chose 15% as a threshold for composition bias (Figure S2). Using this 15% threshold, our gold standard list contains 105 (62.9%) acidic (net charge < -3), 12 (7.19%) glutamine-rich (Q-rich), and 30 (18.0%) proline-rich (P-rich) activation domains.”

3) Net charge of acidic ADs from GSL doesn't look very different from all ADs (Fig. 3C vs Fig. S7B). Doesn't it suggest that 15% threshold is not an ideal criterion for categorizing ADs?

These two distributions look very similar for three reasons. First, greater than two-thirds of activation domains on all of our lists are acidic. Second, there is a plotting artifact: tiles from acidic activation domains show a broader range of acidity than tiles from other types of domains, spreading them out more. For instance, the Q-rich activation domains in Figure S5 pile up between 0 and -2. Third, there is substantial overlap between the acidic and S-rich activation domains. We consider the composition-based classification a historical heuristic. Going forward, we aspire to develop a functional classification system, but we are still in the early stages of this project.

4) Fig 1E - why are random negative controls included in the plot? It seems unnecessary to include them here and leaving them out would increase the overlap between the lists.

Thank you for this suggestion. This is a great idea. Please see the revised Figure 1E and Figure 1F.

5) Fig S3 should be eliminated as the authors state that this analysis is not useful.

When we shared drafts of the manuscript with colleagues, several were curious about the comparison between normalized and non-normalized distributions, so we will leave Figure S3 in place.

6) Pg 8, top: there is no reference to a figure or table with the ser-enriched ADs. In contrast to this statement Fig S6 seems to show no such enrichment.

Thank you for pointing out that the Serine enrichment is hard to see in Figure S6. We have added a panel to Figure S6. Showing Serine enrichment is a difficult and controversial practice because the human proteome is already Serine rich. Choosing different normalization methods provides opposite results (DeiRosso *et al.* 2023). We did not dive into this controversy because it was not central to our message.

7) Table 2: error in table, value should be 0.098, not .98.

Thank you.

8) Table 2 is confusing. I would like to know the fraction of GSL sequences are correctly predicted - not the percent overlap between all predictions and the GSL.

We have added this analysis as a new column in Table 2.

9) Why is Fig 4A,B plotted on a different x axis scale compared with 4C?

Thank you for pointing out this mistake.

10) I suggest that it would be easier to understand the explanation of new boundaries if it is shifted to the "Dissecting the mechanistic predictor" section.

Thank you for this suggestion. We reorganized that section.

References

Avsec Ž., M. Weilert, A. Shrikumar, S. Krueger, A. Alexandari, *et al.*, 2021 Base-resolution models of transcription-factor binding reveal soft motif syntax. *Nat. Genet.* 53: 354–366.

DelRosso N., J. Tycko, P. Suzuki, C. Andrews, Aradhana, *et al.*, 2023 Large-scale mapping and mutagenesis of human transcriptional effector domains. *Nature*.
<https://doi.org/10.1038/s41586-023-05906-y>

Mahatma S., L. Van den Broeck, N. Morffy, M. V. Staller, L. C. Strader, *et al.*, 2023 Prediction and functional characterization of transcriptional activation domains, pp. 1–6 in *2023 57th Annual Conference on Information Sciences and Systems (CISS)*,.

Staller M. V., E. Ramirez, S. R. Kotha, A. S. Holehouse, R. V. Pappu, *et al.*, 2022 Directed mutational scanning reveals a balance between acidic and hydrophobic residues in strong human activation domains. *Cell Syst* 13: 334–345.e5.

Tycko J., N. DelRosso, G. T. Hess, Aradhana, A. Banerjee, *et al.*, 2020 High-Throughput Discovery and Characterization of Human Transcriptional Effectors. *Cell* 183: 2020–2035.e16.

June 13, 2023

GENETICS-2023-306181

Clusters of acidic and hydrophobic residues can predict acidic transcriptional activation domains from protein sequence

Dear Dr. Staller:

Three experts in the field have reviewed your manuscript, and I have read it as well. I am pleased to inform you that, with minor revisions, it is potentially suitable for publication in GENETICS. One reviewer has comments and concerns that need to be addressed in a revised manuscript or in a rebuttal to me. You can read all reviews at the end of this email.

Two reviewers (2 and 3) are satisfied with the revision. Reviewer 1 is more critical and this relates to specific metrics about if the new classifier is improved or not, raising issues with how "improvement" is quantified. I would be satisfied with a reasonable attempt to adjust language or make clear statements about how the classifier is improved or not. In some senses, even the exploration of what the classifier means already has value and the paper can go forward. If possible, please attempt to clarify statements about CNN and the combined approach that are referenced by Reviewer 1, if possible.

We look forward to receiving your revised manuscript. Please let the editorial office know approximately how long you expect to need for revisions.

Upon resubmission, please include:

1. A clean version of your manuscript;
2. A marked version of your manuscript in which you highlight significant revisions carried out in response to the major points raised by the editor/reviewers (track changes is acceptable if preferred);
3. A detailed response to the editor's/reviewers' comments and to the concerns listed above. Please reference line numbers in this response to aid the editors.

Additionally, please ensure that your resubmission is formatted for GENETICS.

<https://academic.oup.com/genetics/pages/general-instructions>

Follow this link to submit the revised manuscript: Link Not Available

Sincerely,

Craig Kaplan
Associate Editor
GENETICS

Approved by:
Karen Arndt
Senior Editor
GENETICS

Reviewer #1 (Comments for the Authors (Required)):

We thank the authors for their responses to our comments. The authors made many improvements to the work to clarify the introduction, addition of table 7, its accompanying paragraphs, and many smaller edits throughout.

While these additions are a step in the right direction, the paper still refers to an improved predictor, but no convincing evidence is provided for this claim. While these additions are a step in the right direction, the paper still claims to present a new model as an improvement to the old model. Additionally, table 4 from the section 'An improved mechanistic predictor' still does not demonstrate an improvement in performance of the new model. While Table 4 shows an increase in the number of predictions that appear on the gold standard and soto list, there is a much larger increase in the number of total predictions. This leaves us with a huge number of false positives, while many of these could be mislabelled true positives, as things are now, this greatly impacts the model's performance for the worse. Specifically, this is shown in the 'proportion of overlapping with ...' columns, though the specific metric shown in these is not labelled, I assume its precision. Also, as we stated before, similar decreases are

also reflected in the percentages shown in figures 5 and 6 when comparing the previous or new model with CNNs. These strongly suggest the old model has better predictive performance than the new model. (As a minor source of confusion, the authors also are inconsistent in referring to the new model as "improved model", "mechanistic model" and sometimes "improved mechanistic model".)

On the other hand, the improved performance demonstrated by the combination of the CNN with either the old or new model is supported by the data presented. However, combining classifiers to yield a better performance is an expected result (this is referred to an "ensemble" approach). The authors argument that this observation indicates that the CNNs are likely based on compositions does not make sense. If the CNN was also using composition, one would predict little improvement for the two composition based classifiers. On the other hand, when classifiers are using different signals, their combination often leads to an improvement. Of course, the CNN could be using a different aspect of composition. But the point is that their observation does not provide evidence as to whether the CNN is using composition or not. (In passing, the authors argument that interpretation approaches cannot be used for protein CNNs is does not make sense. Indeed, Erijman et al show this in figure 4 of their paper.)

Reviewer #2 (Comments for the Authors (Required)):

The revised manuscript clarifies the issues raised by me and other reviewers. I believe it is a valuable contribution to the field and I look forward to future manuscripts from your lab.

Reviewer #3 (Comments for the Authors (Required)):

The authors have revised the manuscript in response to my earlier comments and to those of the other reviewers. I feel that the authors have adequately addressed my comments. I think that this work is valuable to the field as it gives interesting insight into features important for AD function. Even though the performance of the prediction model isn't ideal, I agree with the authors' conclusion that "the mechanistic predictor is valuable because it is simple and interpretable". The results here support and strengthen earlier proposals and provide explanations for why several AD features are important. I recommend publication in Genetics.

Associate Editor Comments:

Point by point Response to Second Round of Reviews

Reviewer #1 (Comments for the Authors (Required)):

We thank the authors for their responses to our comments. The authors made many improvements to the work to clarify the introduction, addition of table 7, its accompanying paragraphs, and many smaller edits throughout .

We are very glad that you agree that the changes have improved the manuscript.

While these additions are a step in the right direction, the paper still refers to an improved predictor, but no convincing evidence is provided for this claim. While these additions are a step in the right direction, the paper still claims to present a new model as an improvement to the old model. Additionally, table 4 from the section 'An improved mechanistic predictor' still does not demonstrate an improvement in performance of the new model. While Table 4 shows an increase in the number of predictions that appear on the gold standard and soto list, there is a much larger increase in the number of total predictions.

Thank you for pointing out that we were not precise with our use of the term 'improved.' We added three sentences to clarify what we mean by 'improved.'

- Pg 4, line 143: "The new model predicted many more activation domains with minimal loss of accuracy."
- Pg 10, line 441: "The primary improvement of this expanded predictor is that it makes many new predictions with small decreases in accuracy (see below)."
- Pg 12, line 548: "Finally, as previously published, the most rigorous assessment of our predictor is experimental validation (Staller et al. 2022). When we tested the 144 predictions from the mechanistic predictor, 72% had detectable activity. This precision of 0.72 is comparable to the maximum of our estimates above (Table 7). The PADDLE CNN achieves a similar level of precision, 70%, in the recent screen for human activation domains (DelRosso et al. 2023). This CNN performance is comparable to our published success rate (Staller et al. 2022) and the new calculated estimates above (Table 7). Together, these analyses show the improved mechanistic predictor identifies many more candidate activation domains with little to no loss of accuracy."

This leaves us with a huge number of false positives, while many of these could be mislabelled true positives, as things are now, this greatly impacts the model's performance for the worse.

We disagree with this statement that there are a large number of false positives in the improved predictions. The best estimate for the false positive rate (1-sensitivity) is lines 3-5 of Table 7, which estimate the sensitivity of the improved predictor to be 0.692-0.734. This false positive rate is inline with the rates of the original predictor (0.72) and PADDLE (0.70). The improved predictor makes many more predictions with little to no loss of sensitivity.

Specifically, this is shown in the 'proportion of overlapping with ...' columns, though the specific metric shown in these is not labelled, I assume its precision.

Thank you for this good suggestion. We have changed two column headings in Table 4 to include "Precision."

Also, as we stated before, similar decreases are also reflected in the percentages shown in figures 5 and 6 when comparing the previous or new model with CNNs. These strongly suggest the old model has better predictive performance than the new model.

It was very important to us to make new predictions because we had already tested all the predictions from the original mechanistic predictor. Based on Table 7 and the experimental validation rate of the original mechanistic predictor (72%), we believe that the new predictor makes more predictions with little to no sacrifice in accuracy (see also above).

(As a minor source of confusion, the authors also are inconsistent in referring to the new model as "improved model", "mechanistic model" and sometimes "improved mechanistic model".)

Thank you for pointing out the inconsistencies in our terminology for the "original predictor" and the "improved mechanistic predictor." We rechecked the document to be more consistent and changed the labels of many entries in Tables 3-6.

On the other hand, the improved performance demonstrated by the combination of the CNN with either the old or new model is supported by the data presented. However, combining classifiers to yield a better performance is an expected result (this is referred to an "ensemble" approach).

For this reviewer point, we believe that the relevant part of the text is the following paragraph (Pg 11):

Why does the combination of the mechanistic predictor improve performance? CNNs are black box models, which makes it difficult to understand the source of their accuracy. ADpred and PADDLE take as inputs primary sequence, predicted secondary structure, and predicted intrinsic disorder, but the models do not tell us which feature, or combination of features, is most important for prediction. For regulatory DNA CNNs, there are emerging tools for extracting mechanistic insight (Avsec *et al.* 2021), but analogous tools for protein sequence have not yet been established. The overlap between the mechanistic predictor and the CNNs suggests that composition, and not grammar, plays a large role in CNN performance.

We had intended this paragraph to be a little speculative and stimulate discussion. This idea is not essential to the paper. We would be happy to cut the paragraph if necessary. In the meantime, we have revised the language to emphasize that we are speculating:

Why does the combination of the mechanistic predictor and the CNNs improve performance? Some of this improvement is likely because each approach is bringing orthogonal information. We also believe that the overlap might be providing some insight into how the CNNs work. CNNs are black box models, which makes it difficult to understand the source of their accuracy. ADpred and PADDLE take as inputs primary sequence, predicted secondary structure, and predicted intrinsic disorder, but the models do not tell us which feature, or combination of features, is most important for prediction. For regulatory DNA CNNs, there are emerging tools for extracting mechanistic insight (Avsec et al. 2021), but analogous tools for interpreting protein sequence models remain limited (Erijman et al. 2020; Mahatma et al. 2023). We speculate that the overlap between the mechanistic predictor and the CNNs suggests that composition plays a substantial role in CNN performance.

The authors argument that this observation indicates that the CNNs are likely based on compositions does not make sense. If the CNN was also using composition, one would predict little improvement for the two composition based classifiers. On the other hand, when classifiers are using different signals, their combination often leads to an improvement.

There are two observations: 1) there is substantial overlap between the mechanistic predictor and the CNNs. 2) Combining the mechanistic predictors with the CNNs increases the precision of the predictions. We agree that this improvement is due to the ensemble approach, which implies that each method is bringing different information.

The mechanistic predictor is only predicting on composition. The CNNs are predicting based on composition and grammar. We have confirmed for ourselves that PADDLE uses more than just composition to make predictions, because shuffling the sequence of an AD leads to a wide distribution of predicted activities (data not shown). This result mirrors experimental results seen by us (Staller et al 2018) and Sanborn et al 2020.

Given the extreme simplicity of the mechanistic predictor, we think the observed overlap between the mechanistic model and the CNNs is very high. The mechanistic predictor overlap with ADpred is 21% and the overlap with PADDLE is 24%, higher than overlap between ADpred and PADDLE (20.5%). This high level of overlap makes us think that roughly a quarter of the predictive power of the CNNs comes from composition. We want to include this speculation that ADpred and PADDLE might be using composition to predict activity to encourage discussion in the literature.

Of course, the CNN could be using a different aspect of composition. But the point is that their observation does not provide evidence as to whether the CNN is using composition or not. **(In passing, the authors argument that interpretation approaches cannot be used for protein CNNs is does not make sense. Indeed, Erijman et al show this in figure 4 of their paper.)**

In our previous response to reviewer comments, we said: "CNNs are black box models that are very difficult to interpret mechanistically." We did not say that interpretation was impossible. Thank you for bringing up the CNN interpretation performed by Erijman et al., we had under-engaged with that analysis in the preparation of this manuscript. The gradient descent

interpretation in Erijman et al. visualizes the residues in specific ADs that contribute most to the prediction. It is interpreting how individual ADs work. It is not interpreting how the model works. To us, this difference is critical: the former lends understanding to how individual ADs function, while the latter would reveal unifying principles. We want to know what the model has learned so that we can understand how ADs work. Using the gradient descent, one would need to interpret ADs and average the signal in a meaningful way. The averaging seems really difficult. From the 33 examples shown in Erijman et al., the model has learned to recognize W's, F's, Y's, E's, and D's as positive signals and R's and K's as negative signals. These signals are precisely the same composition signatures present in our predictor. From these 33 examples, it is clear that ADpred has more signal than just composition, because the W's are taller when they are more C-terminal, and the acidic are shorter when there is run of successive residues.

With collaborators, we have built a new CNN model with a different architecture that uses computed sequence features (e.g. net charge, hydrophobicity, etc) instead of primary sequence. We interpreted this model with SHAP (SHapley Additive exPlanations), SHAP and it was very informative (S. Lundberg and S.-I. Lee, "A Unified Approach to Interpreting Model Predictions," arXiv, 2017, doi: 10.48550/arxiv.1705.07874). We described the model and the first SHAP analysis here: S Mahatma, L Van den Broeck, N Morffy, MV Staller, LC Strader, R Sozzani R. 2023. Prediction and functional characterization of transcriptional activation domains 2023 57th Annual Conference on Information Sciences and Systems (CISS). pp. 1–6. We must emphasize that while the SHAP was really useful for interpreting our sequence feature model, we think it will not be useful for interpreting one-hot encoding models that take as input protein sequences like ADpred or PADDLE.

We have revised the sentence on pg 11, lines 510-512:

For regulatory DNA CNNs, there are emerging tools for extracting mechanistic insight (Avsec et al. 2021), but analogous tools for interpreting protein sequence models remain limited (Erijman et al. 2020; Mahatma et al. 2023).

Reviewer #2 (Comments for the Authors (Required)):

The revised manuscript clarifies the issues raised by me and other reviewers. I believe it is a valuable contribution to the field and I look forward to future manuscripts from your lab.

Reviewer #3 (Comments for the Authors (Required)):

The authors have revised the manuscript in response to my earlier comments and to those of the other reviewers. I feel that the authors have adequately addressed my comments. I think that this work is valuable to the field as it gives interesting insight into features important for AD function. Even though the performance of the prediction model isn't ideal, I agree with the authors' conclusion that "the mechanistic predictor is valuable because it is simple and

interpretable". The results here support and strengthen earlier proposals and provide explanations for why several AD features are important. I recommend publication in Genetics.

June 26, 2023

RE: GENETICS-2023-306271

Dear Dr. Staller:

I am pleased to accept your manuscript entitled "**Clusters of acidic and hydrophobic residues can predict acidic transcriptional activation domains from protein sequence**" for publication in GENETICS, pending minor revision. I had emailed you last week regarding the points below but had not heard back. We are still prepared to accept this pending minor revision and appropriate responses to the below points..

Please submit your revision along with a brief description of how you modified the manuscript in response my comments.

Each and all points of R1 need a rebuttal if possible. I have indicated sentences with asterisks and an added number to indicate points that I am not sure I see specific responses to.

Quoting R1: "While these additions are a step in the right direction, the paper still refers to an improved predictor, but no convincing evidence is provided for this claim. While these additions are a step in the right direction, the paper still claims to present a new model as an improvement to the old model. Additionally, table 4 from the section 'An improved mechanistic predictor' still does not demonstrate an improvement in performance of the new model. While Table 4 shows an increase in the number of predictions that appear on the gold standard and soto list, there is a much larger increase in the number of total predictions. ****(1) This leaves us with a huge number of false positives, while many of these could be mislabelled true positives, as things are now, this greatly impacts the model's performance for the worse. **** Specifically, this is shown in the 'proportion of overlapping with ...' columns, though the specific metric shown in these is not labelled, I assume its precision. Also, as we stated before, similar decreases are also reflected in the percentages shown in figures 5 and 6 when comparing the previous or new model with CNNs. These strongly suggest the old model has better predictive performance than the new model. (As a minor source of confusion, the authors also are inconsistent in referring to the new model as "improved model", "mechanistic model" and sometimes "improved mechanistic model".)

On the other hand, the improved performance demonstrated by the combination of the CNN with either the old or new model is supported by the data presented. However, combining classifiers to yield a better performance is an expected result (this is referred to an "ensemble" approach). *****(2) The authors argument that this observation indicates that the CNNs are likely based on compositions does not make sense. If the CNN was also using composition, one would predict little improvement for the two composition based classifiers. On the other hand, when classifiers are using different signals, their combination often leads to an improvement. **** Of course, the CNN could be using a different aspect of composition. But the point is that their observation does not provide evidence as to whether the CNN is using composition or not. ****(3)(In passing, the authors argument that interpretation approaches cannot be used for protein CNNs is does not make sense. Indeed, Erijman et al show this in figure 4 of their paper.)****"

For point 1, if the reviewer is incorrect in assumption that there is a huge false positive rate, this should be indicated.

For point 2, I think your text indicates that there is some similarity in predictors, though the combined predictor still does better, is this the case? And the similarity is why you are suggesting that the CNN might be using composition? Potentially this could be even more explicit.

For point 3, it seems that R1 is specifically indicating that CNNs are beginning to be interpreted. Are you suggesting that R1s example does not represent an emerging tool like those for DNA CNNs? I want to make sure that this point is addressed or that you specifically indicate how your revision addresses it.

I expect you should be able to submit a revised manuscript within 30 days. A suitably revised manuscript will be acceptable for publication; I don't expect to send it out for review.

Thank you for submitting this story to Genetics.

Sincerely,

Craig Kaplan
Associate Editor
GENETICS

Approved by:
Karen Arndt
Senior Editor
GENETICS

Reviewer comments:

Associate Editor comments:

Point by point Response to Second Round of Reviews

Reviewer #1 (Comments for the Authors (Required)):

We thank the authors for their responses to our comments. The authors made many improvements to the work to clarify the introduction, addition of table 7, its accompanying paragraphs, and many smaller edits throughout .

We are very glad that you agree that the changes have improved the manuscript.

While these additions are a step in the right direction, the paper still refers to an improved predictor, but no convincing evidence is provided for this claim. While these additions are a step in the right direction, the paper still claims to present a new model as an improvement to the old model. Additionally, table 4 from the section 'An improved mechanistic predictor' still does not demonstrate an improvement in performance of the new model. While Table 4 shows an increase in the number of predictions that appear on the gold standard and soto list, there is a much larger increase in the number of total predictions.

Thank you for pointing out that we were not precise with our use of the term 'improved.' We added three sentences to clarify what we mean by 'improved.'

- Pg 4, line 143: "The new model predicted many more activation domains with minimal loss of accuracy."
- Pg 10, line 441: "The primary improvement of this expanded predictor is that it makes many new predictions with small decreases in accuracy (see below)."
- Pg 12, line 548: "Finally, as previously published, the most rigorous assessment of our predictor is experimental validation (Staller et al. 2022). When we tested the 144 predictions from the mechanistic predictor, 72% had detectable activity. This precision of 0.72 is comparable to the maximum of our estimates above (Table 7). The PADDLE CNN achieves a similar level of precision, 70%, in the recent screen for human activation domains (DelRosso et al. 2023). This CNN performance is comparable to our published success rate (Staller et al. 2022) and the new calculated estimates above (Table 7). Together, these analyses show the improved mechanistic predictor identifies many more candidate activation domains with little to no loss of accuracy."

This leaves us with a huge number of false positives, while many of these could be mislabelled true positives, as things are now, this greatly impacts the model's performance for the worse.

We disagree with this statement that there are a large number of false positives in the improved predictions. The best estimate for the false positive rate (1-sensitivity) is lines 3-5 of Table 7, which estimate the sensitivity of the improved predictor to be 0.692-0.734. This false positive rate is inline with the rates of the original predictor (0.72) and PADDLE (0.70). The improved predictor makes many more predictions with little to no loss of sensitivity.

Specifically, this is shown in the 'proportion of overlapping with ...' columns, though the specific metric shown in these is not labelled, I assume its precision.

Thank you for this good suggestion. We have changed two column headings in Table 4 to include "Precision."

Also, as we stated before, similar decreases are also reflected in the percentages shown in figures 5 and 6 when comparing the previous or new model with CNNs. These strongly suggest the old model has better predictive performance than the new model.

It was very important to us to make new predictions because we had already tested all the predictions from the original mechanistic predictor. Based on Table 7 and the experimental validation rate of the original mechanistic predictor (72%), we believe that the new predictor makes more predictions with little to no sacrifice in accuracy (see also above).

(As a minor source of confusion, the authors also are inconsistent in referring to the new model as "improved model", "mechanistic model" and sometimes "improved mechanistic model".)

Thank you for pointing out the inconsistencies in our terminology for the "original predictor" and the "improved mechanistic predictor." We rechecked the document to be more consistent and changed the labels of many entries in Tables 3-6.

On the other hand, the improved performance demonstrated by the combination of the CNN with either the old or new model is supported by the data presented. However, combining classifiers to yield a better performance is an expected result (this is referred to an "ensemble" approach).

For this reviewer point, we believe that the relevant part of the text is the following paragraph (Pg 11):

Why does the combination of the mechanistic predictor improve performance? CNNs are black box models, which makes it difficult to understand the source of their accuracy. ADpred and PADDLE take as inputs primary sequence, predicted secondary structure, and predicted intrinsic disorder, but the models do not tell us which feature, or combination of features, is most important for prediction. For regulatory DNA CNNs, there are emerging tools for extracting mechanistic insight (Avsec *et al.* 2021), but analogous tools for protein sequence have not yet been established. The overlap between the mechanistic predictor and the CNNs suggests that composition, and not grammar, plays a large role in CNN performance.

We had intended this paragraph to be a little speculative and stimulate discussion. This idea is not essential to the paper. We would be happy to cut the paragraph if necessary. In the meantime, we have revised the language to emphasize that we are speculating:

Why does the combination of the mechanistic predictor and the CNNs improve performance? Some of this improvement is likely because each approach is bringing orthogonal information. We also believe that the overlap might be providing some insight into how the CNNs work. CNNs are black box models, which makes it difficult to understand the source of their accuracy. ADpred and PADDLE take as inputs primary sequence, predicted secondary structure, and predicted intrinsic disorder, but the models do not tell us which feature, or combination of features, is most important for prediction. For regulatory DNA CNNs, there are emerging tools for extracting mechanistic insight (Avsec et al. 2021), but analogous tools for interpreting protein sequence models remain limited (Erijman et al. 2020; Mahatma et al. 2023). We speculate that the overlap between the mechanistic predictor and the CNNs suggests that composition plays a substantial role in CNN performance.

The authors argument that this observation indicates that the CNNs are likely based on compositions does not make sense. If the CNN was also using composition, one would predict little improvement for the two composition based classifiers. On the other hand, when classifiers are using different signals, their combination often leads to an improvement.

There are two observations: 1) there is substantial overlap between the mechanistic predictor and the CNNs. 2) Combining the mechanistic predictors with the CNNs increases the precision of the predictions. We agree that this improvement is due to the ensemble approach, which implies that each method is bringing different information.

The mechanistic predictor is only predicting on composition. The CNNs are predicting based on composition and grammar. We have confirmed for ourselves that PADDLE uses more than just composition to make predictions, because shuffling the sequence of an AD leads to a wide distribution of predicted activities (data not shown). This result mirrors experimental results seen by us (Staller et al 2018) and Sanborn et al 2020.

Given the extreme simplicity of the mechanistic predictor, we think the observed overlap between the mechanistic model and the CNNs is very high. The mechanistic predictor overlap with ADpred is 21% and the overlap with PADDLE is 24%, higher than overlap between ADpred and PADDLE (20.5%). This high level of overlap makes us think that roughly a quarter of the predictive power of the CNNs comes from composition. We want to include this speculation that ADpred and PADDLE might be using composition to predict activity to encourage discussion in the literature.

Of course, the CNN could be using a different aspect of composition. But the point is that their observation does not provide evidence as to whether the CNN is using composition or not. **(In passing, the authors argument that interpretation approaches cannot be used for protein CNNs is does not make sense. Indeed, Erijman et al show this in figure 4 of their paper.)**

In our previous response to reviewer comments, we said: "CNNs are black box models that are very difficult to interpret mechanistically." We did not say that interpretation was impossible. Thank you for bringing up the CNN interpretation performed by Erijman et al., we had under-engaged with that analysis in the preparation of this manuscript. The gradient descent

interpretation in Erijman et al. visualizes the residues in specific ADs that contribute most to the prediction. It is interpreting how individual ADs work. It is not interpreting how the model works. To us, this difference is critical: the former lends understanding to how individual ADs function, while the latter would reveal unifying principles. We want to know what the model has learned so that we can understand how ADs work. Using the gradient descent, one would need to interpret ADs and average the signal in a meaningful way. The averaging seems really difficult. From the 33 examples shown in Erijman et al., the model has learned to recognize W's, F's, Y's, E's, and D's as positive signals and R's and K's as negative signals. These signals are precisely the same composition signatures present in our predictor. From these 33 examples, it is clear that ADpred has more signal than just composition, because the W's are taller when they are more C-terminal, and the acidic are shorter when there is run of successive residues.

With collaborators, we have built a new CNN model with a different architecture that uses computed sequence features (e.g. net charge, hydrophobicity, etc) instead of primary sequence. We interpreted this model with SHAP (SHapley Additive exPlanations), SHAP and it was very informative (S. Lundberg and S.-I. Lee, "A Unified Approach to Interpreting Model Predictions," arXiv, 2017, doi: 10.48550/arxiv.1705.07874). We described the model and the first SHAP analysis here: S Mahatma, L Van den Broeck, N Morffy, MV Staller, LC Strader, R Sozzani R. 2023. Prediction and functional characterization of transcriptional activation domains 2023 57th Annual Conference on Information Sciences and Systems (CISS). pp. 1–6. We must emphasize that while the SHAP was really useful for interpreting our sequence feature model, we think it will not be useful for interpreting one-hot encoding models that take as input protein sequences like ADpred or PADDLE.

We have revised the sentence on pg 11, lines 510-512:

For regulatory DNA CNNs, there are emerging tools for extracting mechanistic insight (Avsec et al. 2021), but analogous tools for interpreting protein sequence models remain limited (Erijman et al. 2020; Mahatma et al. 2023).

Reviewer #2 (Comments for the Authors (Required)):

The revised manuscript clarifies the issues raised by me and other reviewers. I believe it is a valuable contribution to the field and I look forward to future manuscripts from your lab.

Reviewer #3 (Comments for the Authors (Required)):

The authors have revised the manuscript in response to my earlier comments and to those of the other reviewers. I feel that the authors have adequately addressed my comments. I think that this work is valuable to the field as it gives interesting insight into features important for AD function. Even though the performance of the prediction model isn't ideal, I agree with the authors' conclusion that "the mechanistic predictor is valuable because it is simple and

interpretable". The results here support and strengthen earlier proposals and provide explanations for why several AD features are important. I recommend publication in Genetics.

June 30, 2023

RE: GENETICS-2023-306271R1

Dr. Max Valentín Staller
University of California Berkeley
Center for Computational Biology
16 Barker Hall
Berkeley, California 94720

Dear Dr. Staller:

Congratulations! We are delighted to inform you that your manuscript entitled "**Clusters of acidic and hydrophobic residues can predict acidic transcriptional activation domains from protein sequence**" is acceptable for publication in GENETICS. Many thanks for submitting your research to the journal.

To Proceed to Production:

1. Format your article according to GENETICS style, as discussed at <https://academic.oup.com/genetics/pages/general-instructions>, and upload your final files at <https://genetics.msubmit.net>.
2. Your manuscript will be published as-is (unedited-as submitted, reviewed, and accepted) at the GENETICS website as an Advanced Access article and deposited into PubMed shortly after receipt of source files and the completed license to publish. Please notify sourcefiles@thegsajournals.org if you do not wish to publish your article via Advanced Access.
3. We invite you to submit an original color figure related to your paper for consideration as cover art. Please email your submission to the editorial office or upload it with your final files. You can submit a small-sized image for evaluation, and if selected, the final image must be a TIFF file 2513px wide by 3263px high (8.375 by 10.875 inches; resolution of 600ppi). Please avoid graphs and small type.

If you have any questions or encounter any problems while uploading your accepted manuscript files, please email the editorial office at sourcefiles@thegsajournals.org.

Sincerely,

Craig Kaplan
Associate Editor
GENETICS

Approved by:
Karen Arndt
Senior Editor
GENETICS

note: Please add jnls.author.support@oup.com and genetics.oup@kwglobal.com (or the domains @oup.com and @kwglobal.com) to your email program's "safe senders" list. You will be contacted by both at various points during the production process.

Review comments (if applicable):